# On Extending Direct Preference Optimization to Accommodate Ties

**Jinghong Chen, Guangyu Yang, Weizhe Lin, Jingbiao Mei, Chenxu Lyu, Bill Byrne**
Department of Engineering
University of Cambridge
Cambridge, United Kingdom CB2 1PZ
{jc2124, gy266, wl356, jm2245, cl927, wjb31}@cam.ac.uk

## Abstract

We derive and investigate two DPO variants that explicitly model the possibility of declaring a tie in pair-wise comparisons. We replace the Bradley-Terry model in DPO with two well-known modeling extensions, by Rao and Kupper and by Davidson, that assign probability to ties as alternatives to clear preferences. Our experiments in neural machine translation and summarization show that explicitly labeled ties can be added to the datasets for these DPO variants without the degradation in task performance that is observed when the same tied pairs are presented to DPO. We find empirically that the inclusion of ties leads to stronger regularization with respect to the reference policy as measured by KL divergence, and we see this even for DPO in its original form. We provide a theoretical explanation for this regularization effect using ideal DPO policy theory. We further show performance improvements over DPO in translation and mathematical reasoning using our DPO variants. We find it can be beneficial to include ties in preference optimization rather than simply discard them, as is done in common practice.

## 1 Introduction

The original formulation of DPO [Rafailov et al., 2023] does not allow for ties. DPO requires training data consisting of paired options, $y_w \succ y_l$, and each of these pairs should represent a clear preference in judgment with no ambiguity as to which is the winner and which is the loser. From this data, the DPO learning procedure encourages the underlying policy to prefer $y_w$ over $y_l$. This formulation does not allow for any ambiguity or uncertainty in the comparison of the paired examples in the training data.

This certainty is not easy to achieve in practice. A common approach is simply to discard data. Dubey et al. [2024, Sec. 4.2.1] apply DPO in post-training of Llama 3 models and note that for "DPO, we use samples that are labeled as the chosen response being significantly better or better than the rejected counterpart for training and discard samples with similar responses." Similarly, Qwen2 developers [Yang et al., 2024a, Sec. 4.3] "sample multiple responses from the current policy model, and the reward model selects the most and the least preferred responses, forming preference pairs that are used for DPO." Over-generation followed by aggressive selection is effective in producing the strongly ordered judgments needed for DPO. However the process appears wasteful: many potentially useful, and expensively collected, preference judgments are discarded simply because they are ties. As Rao and Kupper [1967] note: "any model which does not allow for the possibility of ties is not making full use of the information contained in the no-preference class."

Motivated by this, we investigate DPO variants that can incorporate ties. We replace the Bradley-Terry preference model at the heart of DPO by two well-known extensions by Rao and Kupper [1967] and by Davidson [1970] that explicitly assign probability to tied judgments alongside winners and

losers. Since these models are generalizations of the Bradley-Terry model, we find that they are readily incorporated into the DPO modeling framework. In experiments in neural machine translation and summarization, we find that ties can be added to the datasets for these DPO variants without the degradation in task performance that results from adding ties to the original DPO. We also observe improved regularization, in reduced KL-divergence to the reference policy, by adding ties. We provide a theoretical explanation based on ideal DPO policy theory by Chen et al. [2024]. We further show DPO-RK and DPO-D improves performance over strong DPO baselines in translation and mathematical reasoning by including previously discarded data as tied pairs. These findings suggest it can be beneficial to incorporate ties in preference optimization rather than simply discard them, as is done in common practice.

## 2 Methodology

### 2.1 DPO and the Bradley-Terry Preference Distribution

The Bradley-Terry model assigns probability that an item $y_i$ will be preferred to item $y_j$ in terms of their 'strength' parameters $\lambda$. In the RLHF setting, strengths are expressed as rewards $r$, $\lambda = e^r$ [Eq. 1] Rafailov et al. [2023], so that the preference distribution for item $i$ over item $j$ depends on the difference in their rewards, $d_{ij} = r_i - r_j$

$$p^{BT}(y_i \succ y_j) = \frac{\lambda_i}{\lambda_i + \lambda_j} = \frac{e^{r_i}}{e^{r_i} + e^{r_j}} = \sigma(r_i - r_j) = \sigma(d_{ij}) \tag{1}$$

One of the enabling observations made by Rafailov et al. [2023] is that when a policy $\pi_\theta$ is sought to maximize the KL-regularized objective $\max_{\pi_\theta} \mathbb{E}\left[r(x, y)\right] - \beta\, D(\pi_\theta(y|x) \,||\, \pi_{ref}(y|x))$, the reward associated with the policy has the form $r_\theta(x, y) = \beta \log \frac{\pi_\theta(y|x)}{\pi_{ref}(y|x)} + \beta \log Z_\theta(x)$. This allows expressing the difference in rewards between hypotheses $y_w$ and $y_l$ under a parameterized policy $\pi_\theta$ as the reward margin

$$d_\theta(x, y_w, y_l) = r_\theta(x, y_w) - r_\theta(x, y_l) = \beta \log \frac{\pi_\theta(y_w|x)}{\pi_{ref}(y_w|x)} - \beta \log \frac{\pi_\theta(y_l|x)}{\pi_{ref}(y_l|x)} \tag{2}$$

so that the corresponding Bradley-Terry probability that item $y_w$ beats item $y_l$ is

$$p_\theta^{BT}(y_w \succ_x y_l) = \sigma(d_\theta(x, y_w, y_l)) = \sigma\big(\beta \log \frac{\pi_\theta(y_w|x)}{\pi_{ref}(y_w|x)} - \beta \log \frac{\pi_\theta(y_l|x)}{\pi_{ref}(y_l|x)}\big) \tag{3}$$

The DPO policy objective [Eq. 7]Rafailov et al. [2023] follows by incorporating the parameterized form of the preference distribution into a maximum likelihood objective

$$\mathcal{L}_{DPO}(\pi_\theta; \pi_{ref}) = -\mathbb{E}_{x, y_w, y_l} \log p_\theta(y_w \succ_x y_l) \tag{4}$$

$$= -\mathbb{E}_{x, y_w, y_l} \log \sigma\big(\beta \log \frac{\pi_\theta(y_w|x)}{\pi_{ref}(y_w|x)} - \beta \log \frac{\pi_\theta(y_l|x)}{\pi_{ref}(y_l|x)}\big) \tag{5}$$

We note that Eq. 2 follows from the regularized risk optimization [Rafailov et al., 2023, A.1]. It does not rely on any assumption that limits its use to the Bradley-Terry model.

### 2.2 Bradley-Terry Extensions that Accommodate Ties

An observed weakness of the Bradley-Terry model is that it does not allow for ties. Unless two items have exactly the same strengths (so that $d_{ij} = 0$), the model always assigns a higher probability of winning to the stronger item. This may be reasonable if one item is much stronger than the other, but when items are relatively comparable it may be desirable to allow some probability for tied outcomes.

The Rao-Kupper [Rao and Kupper, 1967] model assigns win and tie probabilities as:

$$p^{RK}(y_i \succ y_j) = \frac{\lambda_i}{\lambda_i + \nu_{RK}\lambda_j} \qquad\qquad y_i \text{ beats } y_j \tag{6}$$

$$p^{RK}(y_i \sim y_j) = \frac{(\nu_{RK}^2 - 1)\lambda_i\lambda_j}{(\lambda_i + \nu_{RK}\lambda_j)(\lambda_j + \nu_{RK}\lambda_i)} \qquad\qquad y_i \text{ ties } y_j \tag{7}$$

while the Davidson [Davidson, 1970] model assigns win and tie probabilities as:

$$p^D(y_i \succ y_j) = \frac{\lambda_i}{\lambda_i + \lambda_j + 2\nu_D \sqrt{\lambda_i \lambda_j}} \qquad \qquad y_i \text{ beats } y_j \qquad (8)$$

$$p^D(y_i \sim y_j) = \frac{2\nu_D \sqrt{\lambda_i \lambda_j}}{\lambda_i + \lambda_j + 2\nu_D \sqrt{\lambda_i \lambda_j}} \qquad \qquad y_i \text{ and } y_j \text{ tie} \qquad (9)$$

The probabilities of the three outcomes sum to one for both of these Bradley-Terry extensions: $p(y_i \succ y_j) + p(y_j \succ y_i) + p(y_i \sim y_j) = 1$. For both models, $p(y_i \sim y_j) = p(y_j \sim y_i)$ and $p(y_i \sim y_j)$ tends towards 0 if $\lambda_j \gg \lambda_i$. Both variants have parameters $\nu$ that control how much probability is allocated to ties. Apart from $\nu_{RK} = 1$ or $\nu_D = 0$, when both variants agree with Bradley-Terry, some probability is reserved for tied outcomes.

The Rao-Kupper and Davidson models arise from different considerations. Rao and Kupper [1967] begin with the formulation $p^{BT}(y_i \succ y_j) = \frac{1}{4} \int_{-(r_i - r_j)}^{\infty} \text{sech}^2(y/2) dy$ [Bradley, 1953] and note its sensitivity to the difference in values $r_i - r_j$. They note that some judges "may not be able to express any real preference" in paired-comparisons if their "sense of perception is not sharp enough" to detect small differences. They reason that a "threshold of sensory perception" is needed such that if the observed difference is less than the threshold, a judge declares a tie. They introduce the sensitivity threshold $\alpha_{RK}$ as follows, $p^{RK}(y_i \succ y_j) = \frac{1}{4} \int_{-(r_i - r_j) + \alpha_{RK}}^{\infty} \text{sech}^2(y/2) dy$, and Eqs. 6 and 7 follow for $\nu_{RK} = e^{\alpha_{RK}}$.

Davidson [1970] starts from Luce's "choice axiom" [Luce, 1959a] which states that a complete system of choice probabilities should satisfy $p(y_i \succ y_j)/p(y_j \succ y_i) = \lambda_i/\lambda_j$, which the Rao-Kupper model fails to do. Davidson [1970] observes that it is desirable for the probability of a tie to "be proportional to the geometric mean of the probabilities of preference". Adding this requirement $p(y_i \sim y_j) = 2\nu_D \sqrt{p(y_i \succ y_y)p(y_j \succ y_i)}$ to the choice axioms yields Eqs. 8 and 9 as a preference model that allows for ties and also satisfies the choice axiom.

The Rao-Kupper win and tie probabilities can be written in a form more useful for DPO (Appendix B.1), with $\nu_{RK} = e^{\alpha_{RK}}$, as

$$p_\theta^{RK}(y_w \succ_x y_l) = \sigma(d_\theta(x, y_w, y_l) - \alpha_{RK}) \qquad (10)$$

$$p_\theta^{RK}(y_w \sim_x y_l) = (\nu_{RK}^2 - 1)\,\sigma(-d_\theta(x, y_w, y_l) - \alpha_{RK})\,p_\theta^{RK}(y_w \succ_x y_l) \qquad (11)$$

and the Davidson win and tie probabilities can be written as

$$p_\theta^D(y_w \succ_x y_l) = \frac{1}{1 + e^{-d_\theta(x, y_w, y_l)} + 2\nu_D e^{-d_\theta(x, y_w, y_l)/2}} \qquad (12)$$

$$p_\theta^D(y_w \sim_x y_l) = 2\,\nu_D\,e^{-d_\theta(x, y_w, y_l)/2}\,p_\theta^D(y_w \succ_x y_l) \qquad (13)$$

Although their parametric forms are different, their treatments of wins and ties are similar (Appendix B.1, Fig. 3). For pairs $(x, y_w, y_l)$ treated as wins, higher likelihood is assigned for higher values of the reward margin $d_\theta(x, y_w, y_l)$. For the Rao-Kupper this is particularly clear, in that the Bradley-Terry preference distribution is simply shifted by $\alpha_{RK}$. Conversely, for pairs $(x, y_w, y_l)$ treated as ties, the probability of declaring a tie is high for small reward margins $d_\theta(x, y_w, y_l)$.

**Balancing Wins and Ties.** In the special case of two evenly matched players ($\lambda_i = \lambda_j$), we are interested in the probability of a tie $p(y_i \sim y_j)$ versus a clear win by either player, $p(y_i \succ y_j) + p(y_j \succ y_i)$. It follows that $P_{RK}(\text{tie}) = \frac{\nu_{RK} - 1}{2} P_{RK}(\text{no tie})$ and $P_D(\text{tie}) = \nu_D P_D(\text{no tie})$. This shows that the parameters $\nu$ determine the probability that equally-matched items are judged as tied or not. $\nu$ can be tuned, but in our work, we assume that equally-matched items will tie with a probability of $1/2$ and so we set $\nu_{RK} = 3$ and $\nu_D = 1$.

## 2.3 Incorporating Rao-Kupper and Davidson Models into DPO

We extend the DPO policy objective (Eq. 4) to include a binary flag $t$ to indicate a tie:

$$\mathcal{L}(\pi_\theta; \pi_{ref}) = -\mathbb{E}_{x, y_w, y_l, t=0} \log p_\theta(y_w \succ_x y_l) - \mathbb{E}_{x, y_w, y_l, t=1} \log p_\theta(y_w \sim_x y_l) \qquad (14)$$

where $p_\theta(y_w \succ y_l)$ and $p_\theta(y_w \sim y_l)$ are taken from either the Rao-Kupper model (Eqs. 10, 11 or the Davidson model (Eqs. 12, 13). Note that in Eq. 14 preference pairs in the dataset are unambiguously either wins ($t = 0$) or ties ($t = 1$). The policy objectives for these two DPO variants are:

$$\mathcal{L}_{RK}(\pi_\theta; \pi_{ref}) = -\mathbb{E}_{x,y_w,y_l,t=0}\Big[\log\sigma(d_\theta(x, y_w, y_l) - \alpha_{RK})\Big]$$
$$-\mathbb{E}_{x,y_w,y_l,t=1}\Big[\log\sigma(-d_\theta(x, y_w, y_l) - \alpha_{RK}) + \log\sigma(d_\theta(x, y_w, y_l) - \alpha_{RK}) - \log(\nu_{RK}^2 - 1)\Big]$$
(15)

and

$$\mathcal{L}_D(\pi_\theta; \pi_{ref}) = -\mathbb{E}_{x,y_w,y_l,t=0}\Big[\log\frac{1}{1 + e^{-d_\theta(x,y_w,y_l)} + 2\nu_D e^{-d_\theta(x,y_w,y_l)/2}}\Big]$$
$$-\mathbb{E}_{x,y_w,y_l,t=1}\Big[\log\frac{2\nu_D e^{-d_\theta(x,y_w,y_l)/2}}{1 + e^{-d_\theta(x,y_w,y_l)} + 2\nu_D e^{-d_\theta(x,y_w,y_l)/2}}\Big]$$
(16)

We refer to these DPO variants as DPO-RK and DPO-D. Like DPO, these objectives depend on the policy $\pi_\theta$ through the reward margin $d_\theta(x, y_w, y_l)$ (Eq. 2). Unlike DPO, the training objective Eq. 14 consists of two competing terms. For pairs $(x, y_w, y_l)$ labeled as wins the objective is to find $\pi_\theta$ to increase the reward margin $d_\theta(x, y_w, y_l)$. However, for pairs labeled as ties the objective is to find $\pi_\theta$ to minimize $|d_\theta(x, y_w, y_l)|$. To simultaneously achieve both these objectives, the underlying policy should learn to model both wins and ties.

### 2.3.1 DPO-RK and DPO-D Updates

Rafailov et al. [2023] show that DPO dynamically adjusts the gradient according to how well the preference objective is optimized for each sample

$$\nabla_\theta \log p_\theta^{BT}(y_w \succ_x y_l) = \underbrace{\sigma(-d_\theta(x, y_w, y_l))}_{\substack{\text{higher weight when reward} \\ \text{estimate is wrong}}} \beta\nabla_\theta \log\frac{\pi_\theta(y_w|x)}{\pi_\theta(y_l|x)}$$
(17)

DPO-RK and DPO-D also adjust their gradients dynamically (Appendix B.2). We define the gradient scale factors $\Delta_{win}$ and $\Delta_{tie}$ to illustrate the DPO-RK and DPO-D gradient updates on wins and ties:

$$\nabla \log p_\theta^{RK}(y_w \succ_x y_l) = \underbrace{\sigma(\alpha - d_\theta(x, y_w, y_l))}_{\Delta_{win}^{RK}(d_\theta)} \beta\nabla_\theta \log\frac{\pi_\theta(y_w|x)}{\pi_\theta(y_l|x)}$$
(18)

$$\nabla_\theta \log p_\theta^{RK}(y_w \sim_x y_l) = \underbrace{\Big[\sigma(\alpha - d_\theta(x, y_w, y_l)) - \sigma(\alpha + d_\theta(x, y_w, y_l))\Big]}_{\Delta_{tie}^{RK}(d_\theta)} \beta\nabla_\theta \log\frac{\pi_\theta(y_w|x)}{\pi_\theta(y_l|x)}$$
(19)

$$\nabla_\theta \log p_\theta^D(y_w \succ_x y_l) = \underbrace{\frac{e^{-d_\theta} + \nu e^{-d_\theta/2}}{1 + e^{-d_\theta} + 2\nu e^{-d_\theta/2}}}_{\Delta_{win}^D(d_\theta)} \beta\nabla_\theta \log\frac{\pi_\theta(y_w|x)}{\pi_\theta(y_l|x)}$$
(20)

$$\nabla_\theta \log p_\theta^D(y_w \sim_x y_l) = \underbrace{\Big[\Delta_{win}^D(d_\theta) - \frac{1}{2}\Big]}_{\Delta_{tie}^D(d_\theta)} \beta\nabla_\theta \log\frac{\pi_\theta(y_w|x)}{\pi_\theta(y_l|x)}$$
(21)

$\nabla \log p_\theta(y_w \succ_x y_l)$: For data labeled as wins, the DPO-RK gradient scale factor has the same form as DPO, but shifted by $\alpha_{RK}$ (Fig. 4). DPO-D has a symmetric scale factor that is not as steep as DPO-RK. All three methods work to increase the reward margin $d_\theta(x, y_w, y_l)$.

$\nabla \log p_\theta(y_w \sim_x y_l)$: For data labeled as ties, the DPO-D and DPO-RK gradient scale factors are odd and work to drive $d_\theta(x, y_w, y_l)$ towards zero, although the DPO-RK scale factor is more aggressive. This is a mechanism not present in DPO.

**Intuition for hyper-parameter** $\alpha_{RK}$ **and** $\nu_D$ **:**    We note that DPO-RK and DPO-D each introduces one additional hyper-parameter relative to DPO. In Appendix B we discuss how the Rao-Kupper and the Davidson variants use these hyper-parameters to control the likelihood of a tie. The updates on ties above (Eq.19 and 21) offer another intuition for these hyper-parameters: $\alpha_{RK}$ and $\nu_D$ control the width of the band in reward margin where there is little gradient contributions from tied pairs whose difference in reward falls within the band. However, for tied pairs whose difference in reward fall outside the band, the gradient updates work to reduce the margin. We find experimentally that performance is not sensitive to the choice of $\alpha_{RK}$ and $\nu_D$ so long as they are in sensible range (Appendix F.1) and so keep to $\alpha_{RK} = \log 3$ and $\nu_D = 1$ as motivated in Sec.2.2.

## 3    Experiments in Adding Ties to DPO, DPO-RK, DPO-D

DPO in its original formulation relies on a static dataset of comparisons $\mathcal{D} = \{x^{(i)}, y_w^{(i)}, y_l^{(i)}\}_{i=1}^N$ where $y_w^{(i)}$ and $y_l^{(i)}$ are preferred and dispreferred responses to a prompt $x^{(i)}$ [Rafailov et al., 2023]. These preferences are assumed to be sampled from some latent reward model and we refer to this dataset as **Clear Preference Pairs** (**CPs**, for short) because they are typically selected to reflect a clear preference between winner and loser as assessed either by human judges or by some trusted automatic metric. We distinguish CPs from **Tied Pairs** (**TPs**). TPs also consist of a winner and a loser, but are very similar in quality. Human judges might be less consistent, or have less confidence, in selecting the winner in a tied pair, and automatic metrics will assign more similar or even conflicting quality scores to TPs than to CPs. We study conflicting preferences in Sec.3.5.

### 3.1    Extending Preference Datasets to Include Ties

As noted, DPO datasets typically are constructed to include only CPs. We develop data selection procedures to generate TPs along with CPs so that we can investigate how DPO changes when Tied Pairs are included in the training data. We follow previous work [Wang et al., 2024, Xiong et al., 2024, Liu et al., 2024] to collect preference pairs by sampling and then ranking model responses. We pair the top- and bottom-ranked responses as CPs, and select TPs as pairs that are close in the ranking. For Neural Machine Translation (NMT) on WMT-21 ZH-EN [Akhbardeh et al., 2021] and IWSLT-17 FR-EN [Cettolo et al., 2017], we rank responses by BLEURT, a widely-used reference-based quality metric [Sellam et al., 2020, Freitag et al., 2023]. For Summarization on TL;DR [Stiennon et al., 2020], we rank responses using the implicit reward function learned by DPO itself, without an external reward model. Appendix E gives experiment details. Studies of these selection strategies can be found in Appendix F.7 and F.8.

### 3.2    Adding Ties to DPO - The Regularizing Effects of Ties

Following prior work [Rafailov et al., 2023, Amini et al., 2024a, Park et al., 2024], we evaluate DPO in terms of task performance versus KL divergence to the reference policy. For each of the three tasks we form two training sets: CP, which contains the Clear Preference Pairs; and CP+TP, which contains both the Clear Preference Pairs and the Tied Pairs. We refer to DPO training on these sets as DPO(CP) and DPO(CP+TP). We note IPO [Azar et al., 2024] yields similar results (Appendix F.5).

The obvious conclusion from these experiments (Figure 1) is that including tied pairs in DPO hurts task performance. All best performing systems are obtained by DPO(CP), with DPO(CP+TP) underperforming for nearly all values of KL relative to the reference policy. This performance degradation from including ties is consistent with common practice in the DPO literature which only keeps pairs with clear preference, filtering others to obtain the best-performing system [Yang et al., 2024a, Dubey et al., 2024]. Consistent with this, the TL;DR results show that removing tied pairs from the DPO dataset leads to improved summarization performance, even when ties are identified by a DPO model in an unsupervised manner. However these results also suggest that tied pairs in the DPO datasets can enhance regularization. By this we mean that including TPs causes DPO to find models that are closer to the reference policy as measured by KL divergence. The overall effect of the reduced task performance and more regularization is to shift the frontier 'down and to the left'.

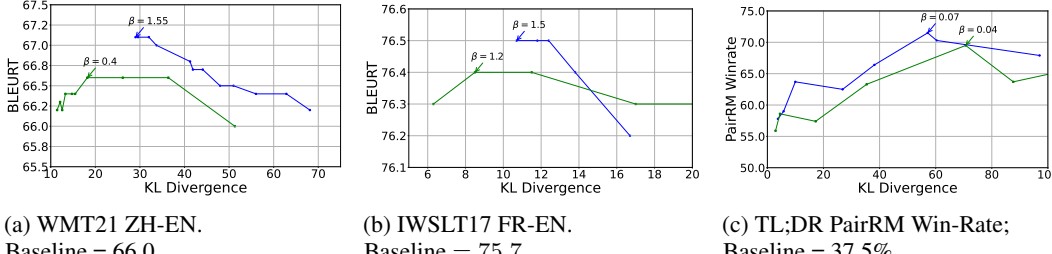

(a) WMT21 ZH-EN.
Baseline = 66.0

(b) IWSLT17 FR-EN.
Baseline = 75.7

(c) TL;DR PairRM Win-Rate;
Baseline = 37.5%

Figure 1: Task Performance vs. KL to the reference policy for DPO systems trained on Clear Preference Pairs (DPO(CP), blue) and on Clear Preference Pairs and Tied Pairs (DPO(CP+TP), green). Task Performance is reported in BLEURT for translation tasks on WMT21 ZH-EN and IWSLT17 FR-EN. Summarization performance is reported on TL;DR in terms of PairRM win-rate against human-written summaries. KL is estimated over 256 test set policy samples; $\beta$ is noted for best performing systems. Full details are in Appendix E.

### 3.3 Explaining the Regularization Effect of Ties via Ideal DPO Policy Theory

Theorem 3.1 of Chen et al. [2024] suggests how these regularization effects might arise. The ideal DPO policy $\pi^*$ should follow (Appendix G):

$$\frac{\pi^*(y_w|x)}{\pi^*(y_l|x)} = \frac{\pi_{\text{ref}}(y_w|x)}{\pi_{\text{ref}}(y_l|x)} \left( \frac{\gamma(x, y_w, y_l)}{1 - \gamma(x, y_w, y_l)} \right)^{1/\beta} \tag{22}$$

where $\gamma(x, y_w, y_l)$ is the true preference probability of $y_w \succ y_l$ under prompt $x$. If we assume that tied pairs have a true preference probability $\gamma(x, y_w, y_l)$ of 0.5, from Equation 22 we have $\frac{\pi^*(y_w|x)}{\pi^*(y_l|x)} = \frac{\pi_{\text{ref}}(y_w|x)}{\pi_{\text{ref}}(y_l|x)}$, where $\pi^*$ is the ideal DPO policy[1]. By this analysis, the ideal DPO model should maintain the same chosen/rejected likelihood ratio as the reference model on tied pairs, and this constraint serves as a form of regularization. In our NMT experiments (Figures 1a, 1b), where half of the pairs are constructed to be ties, the regularization effect is especially pronounced as the DPO model should keep to the reference model likelihood ratio on 50% of the training data. Regularization is less pronounced on TL;DR (Figure 1c) where only 1/8 of the pairs are ties.

Another way to view this is to rearrange Eq 22 as follows:

$$d_\theta^*(x, y_w, y_l) = \beta \left( \log \frac{\pi^*(y_w|x)}{\pi_{ref}(y_w|x)} - \log \frac{\pi^*(y_l|x)}{\pi_{ref}(y_l|x)} \right) = \log \frac{\gamma(x, y_w, y_l)}{1 - \gamma(x, y_w, y_l)} \tag{23}$$

From this it follows that the reward margin on tied pairs should ideally be close to zero, which is a form of regularization. We verify this experimentally in Table 5.

### 3.4 Adding Ties to DPO-RK and DPO-D - Regularization without Performance Degradation

Using the same data as in Sec.3.2, we now evaluate DPO-RK and DPO-D as DPO variants that explicitly model both ties and clear preferences. We use the DPO datasets CP+TP (Sec. 3.2) with the DPO-D and DPO-RK algorithms to produce models DPO-D(CP+TP) and DPO-RK(CP+TP). We follow the protocols of Sec. 3.1 so that results are directly comparable to earlier DPO(CP) and DPO(CP+TP) results. We find that the choice of $\nu$ has only a minor effect on task performance (Appendix F.1). For all experiments we set $\nu^{RK} = 3$ and $\nu^D = 1$ for DPO-RK and DPO-D as described in Sec. 2.2. Training dynamics are provided in Appendix D.

We observe consistent patterns across three datasets: DPO-D(CP+TP) and DPO-RK(CP+TP) do not suffer the same drops in task performance that DPO(CP+TP) exhibits (Fig. 2, orange and purple vs. green). Both reach similar levels of task performance to DPO(CP), but do so at smaller KL values than DPO (Fig. 2, orange and purple vs. blue). For a given level of KL to reference policy, DPO-D(CP+TP) and DPO-RK(DP+TP) yield higher task performance than DPO(CP). Compared to DPO as it is usually done, DPO-RK and DPO-D frontiers are shifted leftwards, showing similar task

---

[1] In Appendix G, we show that the ideal policy can also be derived for DPO-D which includes the ideal DPO policy as a special case.

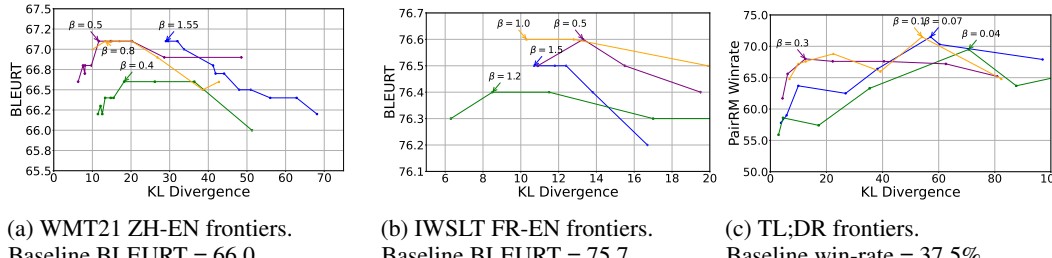

(a) WMT21 ZH-EN frontiers.
Baseline BLEURT = 66.0

(b) IWSLT FR-EN frontiers.
Baseline BLEURT = 75.7

(c) TL;DR frontiers.
Baseline win-rate = 37.5%

Figure 2: KL-Performance frontiers with DPO(CP) in blue, DPO(CP+TP) in green, DPO-RK(CP+TP) in purple, and DPO-D(CP+TP) in orange. Full details in Appendix E.4. For TL;DR, we additionally report win-rate as judged by GPT-4 in Table 13.

performance but stronger regularization. Following the analysis in Sec.3.3, we empirically show the percentage of TPs in the preference dataset is proportional to the regularization effect in Appendix F.4.

### 3.5 Performance Improvement through DPO-RK and DPO-D

In this section, we show two realistic scenarios in machine translation and mathematical reasoning where DPO-RK and DPO-D makes use of otherwise discarded data to achieve performance improvement over strong DPO baselines. We also find DPO-RK and DPO-D leads to overall higher reward accuracy compared to DPO (Appendix C).

#### 3.5.1 Modeling Conflicting Preferences as Ties

In curating preference datasets, it is common to employ multiple human or machine annotators to rank candidate responses. Conflicts between annotators can easily arise. This is the case for the ALMA-R-Preference dataset [Xu et al., 2024], which contains 3 translations sampled from human, GPT-4, and a base model for each source sentence. These translations are ranked by the average of their XCOMET and KIWI-XXL scores, which are reliable reference-less metrics shown to correlate well with human expert judgment [Freitag et al., 2023]. In Xu et al. [2024]'s original work, the pair of translations with the largest average score difference is selected as a CP and the third, middle-scoring, translation is simply discarded. This yields 3070 CPs, one pair for each source sentence.

We propose a simple scheme to treat translations with conflicting preferences as ties. For pairs not selected as CPs, we identify pairs for which XCOMET and KIWI-XXL disagree about which translation is better as pairs of conflicting preferences. For each source sentence where such pair exists, we add one to DPO-RK/DPO-D training as TP, resulting in 2302 TPs. Table 1 shows DPO-RK(CP+TP) and DPO-D(CP+TP) yield better performance across all evaluation metrics compared to DPO(CP). This shows the ability of our DPO variants to learn from pairs of conflicting preferences.

| Model | COMET | KIWI-22 | XCOMET | KIWI-XXL |
|---|---|---|---|---|
| ALMA-7B-LoRA Xu et al. [2024] | 79.78 | 76.81 | 83.94 | 73.65 |
| + DPO(CP) | 79.66 | 77.73 | 88.87 | 74.12 |
| + DPO-RK(CP+TP) | **80.63** | **78.91** | **90.40** | **75.77** |
| + DPO-D(CP+TP) | 80.38 | 78.27 | 90.09 | 75.54 |

Table 1: ZH-EN translation performance on ALMA-R-Preference test set. The best result is reported for DPO(CP), DPO-RK(CP+TP) and DPO-D(CP+TP) over a beta sweep in [0.1, 0.3, 0.5, 0.7, 0.9].

#### 3.5.2 Preserving good behaviour in the reference model through regularization

We have shown that including ties leads to a strong regularization effect with respect to the reference model (Sec.3.2). We now exploit this property to preserve good performance observed in the reference model while leading to stronger overall performance in mathematical reasoning.

Following the DPO-augmented Self-Training (DPO-ST) approach by Wang et al. [2024], we curate preference data by sampling model responses to questions in the GSM8K training set. For 2310 of

7470 questions, we find that the moderately-sized Qwen2.5-3B-Instruct model [Yang et al., 2024b] answers correctly in all of its 8 sampled responses (at temperature 0.7). In DPO-ST, these 2310 questions would be excluded from preference data, even though they are a substantial portion (30.9%) of the training set. We show instead that correct responses to these instances can be treated as ties.

We label CPs and TPs as follows: for training instances where there is at least one correct and one incorrect model response, we randomly choose a correct-versus-incorrect pair as a CP; for examples where all model responses are correct, we randomly choose a pair as a TP. We then conduct DPO training on CP and DPO-RK and DPO-D training on CP+TP with a range of beta values. We also report SimPO and CPO training on the CP set for comparison.

| $\beta$ | SimPO (CP) | CPO (CP) | DPO (CP) | DPO-RK (CP+TP) | DPO-D (CP+TP) |
|---|---|---|---|---|---|
| 0.1 | 82.5% | 82.6% | 76.4% | **83.5%** | 81.7% |
| 0.3 | 81.8% | 83.1% | 83.7% | **84.4%** | 83.2% |
| 0.5 | 81.8% | 83.1% | 83.6% | 83.8% | **84.5%** |
| 0.7 | 81.6% | 82.6% | 83.3% | 83.7% | **84.5%** |
| 1.0 | 82.2% | 83.6% | 83.5% | **84.1%** | 83.7% |

Table 2: GSM8K test set performance with greedy decoding after one-epoch of preference optimization for a range of beta values, evaluated by exact match after "####". Tie-compatible variants yield better performance for every beta value. The base Qwen2.5-3B-Instruct model scores 70.9%. Standard deviation is reported in Appendix E.5.

Table 2 shows that DPO-RK(CP+TP) and DPO-D(CP+TP) achieve better performance than the DPO(CP) model at every beta value. Intuitively, these TPs provide strong regularization signals for examples where the reference model already excels, thereby preserving good behaviour in DPO-RK/DPO-D training. Such a mechanism is not available in the original DPO formulation. To verify this intuition, we define Preservation Rate (PR) as the percentage of questions where the policy continues to answer the question correctly provided that the reference model gives the correct answer under greedy decoding. Lower PR indicates more instances where the answer flips from correct to incorrect after training, which is clearly undesirable. Table 3 shows PR along with KL and overall accuracy on the GSM8K test set. We find that compared to DPO(CP), DPO-RK(CP+TP) and DPO-D(CP+TP) has lower KL and higher PR which leads to higher overall accuracy.

| Model (optimal $\beta$) | KL | Overall Acc% | PR% ↑ |
|---|---|---|---|
| Base model | 0 | 70.9% | 100% |
| DPO(CP), $\beta = 0.3$ | 2.258 | 83.7% | 95.19% |
| DPO-RK(CP+TP), $\beta = 0.3$ | 1.762 | 84.4% | 97.11% |
| DPO-D(CP+TP), $\beta = 0.7$ | 1.465 | **84.5%** | **97.65%** |

Table 3: Preservation Rate (PR%) of the best-performing models.

# 4 Additional Supporting Experiments and Analyses

Due to space constraints, we briefly highlight additional findings and provide pointers to supporting experiments in the appendices.

**DPO-RK and DPO-D yields higher reward accuracy than DPO**   Like DPO, DPO-RK and DPO-D yields an implicit reward function formed by the trained policy and the reference model, which can be used to classify preference pairs. In Table 4, we show that DPO-RK and DPO-D yields overall higher reward accuracy compared to DPO in classifying held-out clear preference pairs and ties. See Appendix C for details.

**The proportion of TPs controls regularization**   We show empirically that the number of ties included in DPO-RK and DPO-D training is proportional to the strength of regularizaion (Table 14). This provides additional support for the regularization effect of ties shown in Sec.3.4.

**Ideal DPO-D policy theory**   Following the ideal DPO policy theory by Chen et al. [2024] used in explaining regularization (Sec.3.3), we derive the ideal DPO-D policy theory starting from the

ternary classification objective that accounts for ties in Appendix G. This can be view as an extension to Chen et al. [2024]'s theory in that the ideal DPO policy emerges as special case.

**Convergence behaviour** In Appendix D, we investigate how reward margins and gradient scale factors (defined in Eqs.17 - 21) evolve when tied pairs are included in training. We find that reward margins on TPs remain close to zero as desired, and that reward margins on CPs rise less sharply and to lower magnitude compared to DPO(CP). An interesting consequence is that, contrary to DPO(CP) training where gradient scale factor decreases to $< 0.05$ after 1/4 epoch and stabilizes, the gradient scale factors remain non-negligible throughout the entirety of training for DPO(CP+TP), DPO-RK(CP+TP), and DPO-D(CP+TP), indicated prolonged learning.

**Qualitative Analyses** Example responses from DPO, DPO-RK, and DPO-D systems are provided and analyzed in Appendix I.

## 5 Related Work

**Related Variants of Direct Preference Optimization** ODPO [Amini et al., 2024b] incorporates preference strength in the objective by introducing an offset parameter. In deriving ODPO, the offset parameter of Amini et al. [2024a, Theorem 3]) plays a role similar to the sensitivity threshold of Rao and Kupper [1967]. We note that the ODPO objective with a fixed offset agrees with our proposed DPO-RK objective restricted to clear preference data, but does not extend to ties. We note DPO-RK is independently proposed by contemporaneous work Guo et al. [2024]. Compared to their work, we additionally introduce DPO-D, propose novel tie selection strategies, identify the regularization effects of ties and provide theoretical explanations. Our experimental setups are complementary: while they primarily focus on general chatbots, we conduct experiments in translation, summarization and mathematical reasoning.

**Frameworks for Pair-wise Preference Optimization** Several works propose theoretical frameworks for understanding general Preference Optimization from which DPO can be obtained as a special case. Dumoulin et al. [2024] formulate learning from pair-wise preference as learning the implicit preference generating distribution of the annotators. In this formalism, DPO is a well-specified model for the implicit preference distribution assuming that the human preference generative process follows the Bradley-Terry model. Our work can be viewed as assuming an annotator preference generating distribution that allows for the outcome of a tie (i.e. the Rao-Kupper or the Davidson model). Tang et al. [2024] propose a generalized approach to deriving offline preference optimization losses through binary classification. In this work, we extend binary classification to ternary classification with the possibility of declaring a tie (Appendix G).

**Pair-wise Comparison Models** Hamilton et al. [2023] review the Bradley-Terry model, including its relation to the logistic distribution [Bradley and Gart, 1962] and Luce choice axiom Luce [1959b]. The Rao-Kupper [Rao and Kupper, 1967] and the Davidson model [David, 1988] are two notable extensions to Bradley-Terry (Sec. 2.2). See the review by David [1988] and bibliography by Davidson and Farquhar [1976]. Modeling ties remains an active research topic in fields such as sport team ranking [Zhou et al., 2022], medical treatments [Gaohong Dong and Vandemeulebroecke, 2020], and chatbots [Ameli et al., 2025].

## 6 Conclusion

We have derived and investigated two tie-compatible DPO variants, DPO-RK and DPO-D, by replacing the Bradley-Terry preference model with the Rao-Kupper and the Davidson models, respectively. Our experiments show that DPO-RK and DPO-D can accommodate tied pairs in preference data without the degradation in task performance that is observed when the same tied pairs are added to the original DPO. We find empirically that the inclusion of ties in preference learning leads to stronger regularization with respect to the reference model and provide theoretical explanations based on ideal DPO policy theory. We further show our DPO variants can improve model performance over DPO by making fuller use of the available data on translation and mathematical reasoning. These findings motivate the use of tied pairs in available preference data as opposed to wastefully discarding them. We discuss limitations in Appendix A.

# 7 Acknowledgement

Jinghong Chen is supported by the Warwick Postgraduate Studentship from Christ's College and the Huawei Hisilicon Studentship for the undertaking of the PhD in Engineering at the University of Cambridge.

Guangyu Yang is supported by Cambridge Commonwealth, European and International Trust for the undertaking of the PhD in Engineering at the University of Cambridge.

Jingbiao Mei is supported by Cambridge Commonwealth, European and International Trust for the undertaking of the PhD in Engineering at the University of Cambridge.

Weizhe Lin is supported by a Research Studentship funded by Toyota Motor Europe (RG92562(24020)) for the undertaking of the PhD in Engineering at the University of Cambridge.

Prof. Bill Byrne holds concurrent appointments as a Professor of Information Engineering at Cambridge University and as an Amazon Scholar. This publication describes work performed at Cambridge University and is not associated with Amazon.

We would also like to thank all the reviewers for their knowledgeable reviews that helped us strengthen the contribution.

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

## A    Limitations

The effect of accommodating ties in preference learning can be further investigated using human-annotated tied pairs. However, at the time of writing, there is no substantial preference dataset with annotated ties; notably, current annotation guidelines are typically written to explicitly exclude ties. We note that this enforcement of win/lose judgments has likely conditioned the generative process of human preference towards the Bradley-Terry model. A meaningful extension of this work would be to assess the effectiveness of DPO-RK and DPO-D on preference datasets where the annotators are asked to identify ties. As explained in Sec 2.2, the hyper-parameter $\nu_{RK}$ and $\nu_D$ can be tuned which would require either grid search or estimation given ground-truth preference/tie probabilities. We find that the choice of $\nu_{RK} = 3$ and $\nu_D = 1$ as motivated in Sec 2.2 works well and we did not need to tune the parameter to obtain good performance. It is likely that better performance and more efficient frontiers can be obtained by tuning $\nu$ to better fit the underlying preference generative process for both DPO-RK and DPO-D. Given our focus on accommodating ties from a modeling perspective, we leave performance optimization to future works concerning applications.

## B    Mathematical Derivations

### B.1    Rao-Kupper and Davidson Preference and Tie Probabilities

We derive the win and tie probabilities as functions of the reward margin $d_\theta(x, y_w, y_l) = r_\theta(x, y_w) - r_\theta(x, y_l)$ (Eq 2) under the Rao-Kupper (Eq 10, 11) and Davidson formulations (Eq 12, 13).

The Rao-Kupper win and tie probabilities can be obtained by substituting $\lambda_w = e^{r_\theta(x,y_w)}$, $\lambda_l = e^{r_\theta(x,y_l)}$ and $\nu_{RK} = e^{\alpha_{RK}}$ into Eq 6 and Eq 7, respectively:

$$
p_\theta^{RK}(y_w \succ y_l) = \frac{\lambda_w}{\lambda_w + \nu_{RK}\lambda_l} = \frac{e^{r_\theta(x,y_w)}}{e^{r_\theta(x,y_w)} + \nu_{RK}e^{r_\theta(x,y_l)}}
$$

$$
= \frac{1}{1 + e^{r_\theta(x,y_l) - r_\theta(x,y_w) + \alpha_{RK}}} = \sigma(d_\theta(x, y_w, y_l) - \alpha_{RK})
$$

$$
p_\theta^{RK}(y_w \sim y_l) = \frac{(\nu_{RK}^2 - 1)\lambda_w\lambda_l}{(\lambda_w + \nu_{RK}\lambda_l)(\lambda_l + \nu_{RK}\lambda_w)}
$$

$$
= \frac{(\nu_{RK}^2 - 1)e^{r_\theta(x,y_w)}e^{r_\theta(x,y_l)}}{(e^{r_\theta(x,y_w)} + \nu_{RK}e^{r_\theta(x,y_l)})(e^{r_\theta(x,y_l)} + \nu_{RK}e^{r_\theta(x,y_w)})}
$$

$$
= (\nu_{RK}^2 - 1)\left(\frac{e^{r_\theta(x,y_l)}}{e^{r_\theta(x,y_l)} + \nu_{RK}e^{r_\theta(x,y_w)}}\right)\left(\frac{e^{r_\theta(x,y_w)}}{e^{r_\theta(x,y_w)} + \nu_{RK}e^{r_\theta(x,y_l)}}\right)
$$

$$
= (\nu_{RK}^2 - 1)\,\sigma(-d_\theta(x, y_w, y_l) - \alpha_{RK})\,\sigma(d_\theta(x, y_w, y_l) - \alpha_{RK})
$$

$$
= (\nu_{RK}^2 - 1)\,\sigma(-d_\theta(x, y_w, y_l) - \alpha_{RK})\,p_\theta^{RK}(y_w \succ y_l)
$$

The Davidson win and tie probabilities can be obtained with the same substitution into Eq 8 and Eq 9, respectively:

$$
p_\theta^D(y_w \succ_x y_l) = \frac{\lambda_w}{\lambda_w + \lambda_l + 2\nu_D\sqrt{\lambda_w\lambda_l}}
$$

$$
= \frac{e^{r_\theta(x,y_w)}}{e^{r_\theta(x,y_w)} + e^{r_\theta(x,y_l)} + 2\nu_D\sqrt{e^{r_\theta(x,y_w) + r_\theta(x,y_l)}}}
$$

$$
= \frac{1}{1 + e^{-d_\theta(x,y_w,y_l)} + 2\nu_De^{-d_\theta(x,y_w,y_l)/2}}
$$

$$
p_\theta^D(y_w \sim_x y_l) = \frac{2\nu_D\sqrt{\lambda_w\lambda_l}}{\lambda_w + \lambda_l + 2\nu_D\sqrt{\lambda_w\lambda_l}}
$$

$$
= (2\nu_D\lambda_w^{-\frac{1}{2}}\lambda_l^{\frac{1}{2}})\frac{\lambda_w}{\lambda_w + \lambda_l + 2\nu_D\sqrt{\lambda_w\lambda_l}}
$$

$$
= 2\,\nu_D\,e^{-\frac{1}{2}(r_\theta(x,y_w) - r_\theta(x,y_l))}\,p_\theta^D(y_w \succ_x y_l)
$$

$$
= 2\,\nu_D\,e^{-d_\theta(x,y_w,y_l)/2}\,p_\theta^D(y_w \succ_x y_l)
$$

In Figure 3 we plot the preference and tie probabilities as a function of reward margin $d_\theta$ under Bradley-Terry (as used in DPO), Rao-Kupper (as used in DPO-RK), and Davidson (as used in DPO-D).

## B.2 Gradients for DPO-RK and DPO-D

The gradients of the Rao-Kupper log probabilities (Eq 18, 19) are as follows. For convenience, we use the short-hand $d_\theta$ for $d_\theta(x, y_w, y_l)$.

$$
\begin{aligned}
\nabla_\theta \log p_\theta^{RK} & (y_w \succ_x y_l) \\
&= \nabla_\theta \log \sigma(d_\theta - \alpha_{RK}) \\
&= \sigma(\alpha_{RK} - d_\theta)\nabla_\theta d_\theta(x, y_w, y_l) \\
&= \underbrace{\sigma(\alpha_{RK} - d_\theta)}_{\Delta_{win}^{RK}(d_\theta)} \Big[\nabla_\theta \log \pi_\theta(y_w|x) - \nabla_\theta \log \pi_\theta(y_l|x)\Big] \\
&= \Delta_{win}^{RK}(d_\theta)\nabla_\theta \log \frac{\pi_\theta(y_w|x)}{\pi_\theta(y_l|x)}
\end{aligned}
$$

$$
\begin{aligned}
\nabla_\theta \log p_\theta^{RK} & (y_w \sim_x y_l) \\
&= \nabla_\theta\Big[\log \sigma(-d_\theta - \alpha_{RK}) + \log \sigma(d_\theta - \alpha_{RK})\Big] \\
&= -\sigma(d_\theta + \alpha_{RK})\nabla_\theta d_\theta + \sigma(-d_\theta + \alpha_{RK})\nabla_\theta d_\theta \\
&= \underbrace{\Big(\sigma(\alpha_{RK} - d_\theta) - \sigma(\alpha_{RK} + d_\theta)\Big)}_{\Delta_{tie}^{RK}(d_\theta)} \nabla_\theta d_\theta \\
&= \Delta_{tie}^{RK}(d_\theta)\nabla_\theta \log \frac{\pi_\theta(y_w|x)}{\pi_\theta(y_l|x)}
\end{aligned}
$$

The gradients of the Davidson log-probabilities (Eq 20, 21) follow similarly.

$$
\begin{aligned}
\nabla_\theta \log p_\theta^{D}(y_w \succ_x y_l) &= \frac{\nabla_\theta p_\theta^{D}(y_w \succ_x y_l)}{p_\theta^{D}(y_w \succ_x y_l)} \\
&= \frac{\nabla_\theta(1 + e^{-d_\theta} + 2\nu e^{-d_\theta/2})^{-1}}{p_\theta^{D}(y_w \succ_x y_l)} \\
&= (-1)\frac{(1 + e^{-d_\theta} + 2\nu e^{-d_\theta/2})^{-2}}{p_\theta^{D}(y_w \succ_x y_l)}(-e^{d_\theta} - \nu e^{d_\theta/2})\nabla_\theta d_\theta \\
&= \frac{p_\theta^{D}(y_w \succ_x y_l)^2}{p_\theta^{D}(y_w \succ_x y_l)}(e^{-d_\theta} + \nu e^{-d_\theta/2})\nabla_\theta d_\theta \\
&= p_\theta^{D}(y_w \succ_x y_l)(e^{-d_\theta} + \nu e^{-d_\theta/2})\nabla_\theta d_\theta \\
&= \underbrace{\frac{e^{-d_\theta} + \nu e^{-d_\theta/2}}{1 + e^{-d_\theta} + 2\nu e^{-d_\theta/2}}}_{\Delta_{win}^{D}(d_\theta)}\nabla_\theta d_\theta \\
&= \Delta_{win}^{D}(d_\theta)\nabla_\theta \log \frac{\pi_\theta(y_w|x)}{\pi_\theta(y_l|x)}
\end{aligned}
$$

$$\nabla_\theta \log p_\theta^D(y_w \sim_x y_l) = \nabla_\theta \log \left(2\nu e^{-d_\theta/2} p_\theta^D(y_w \succ_x y_l)\right)$$

$$= \nabla_\theta \left[\log p_\theta^D(y_w \succ_x y_l) - d_\theta/2\right]$$

$$= \left[\frac{e^{-d_\theta} + \nu e^{-d_\theta/2}}{1 + e^{-d_\theta} + 2\nu e^{-d_\theta/2}} - \frac{1}{2}\right]\nabla_\theta d_\theta$$

$$= \underbrace{\left[\Delta_{win}^D(d_\theta) - \frac{1}{2}\right]}_{\Delta_{tie}^D(d_\theta)}\left[\nabla_\theta \log \pi_\theta(y_w|x) - \nabla_\theta \log \pi_\theta(y_w|x)\right]$$

$$= \Delta_{tie}^D(d_\theta)\nabla_\theta \log \frac{\pi_\theta(y_w|x)}{\pi_\theta(y_l|x)}$$

For illustration, we plot $\Delta_{win}$ and $\Delta_{tie}$ as a function of reward margin $d_\theta$ in Figure 4.

The quantities $\nabla_\theta \mathcal{L}_D(\pi_\theta; \pi_{ref})$ and $\nabla_\theta \mathcal{L}_{RK}(\pi_\theta; \pi_{ref})$ follow by substituting the above results into the gradient of Eq 14

$$\nabla_\theta \mathcal{L}(\pi_\theta; \pi_{ref}) =$$
$$- \nabla_\theta \mathbb{E}_{x,y_w,y_l,t=0} \log p_\theta(y_w \succ_x y_l)$$
$$- \nabla_\theta \mathbb{E}_{x,y_w,y_l,t=1} \log p_\theta(y_w \sim_x y_l) \tag{24}$$

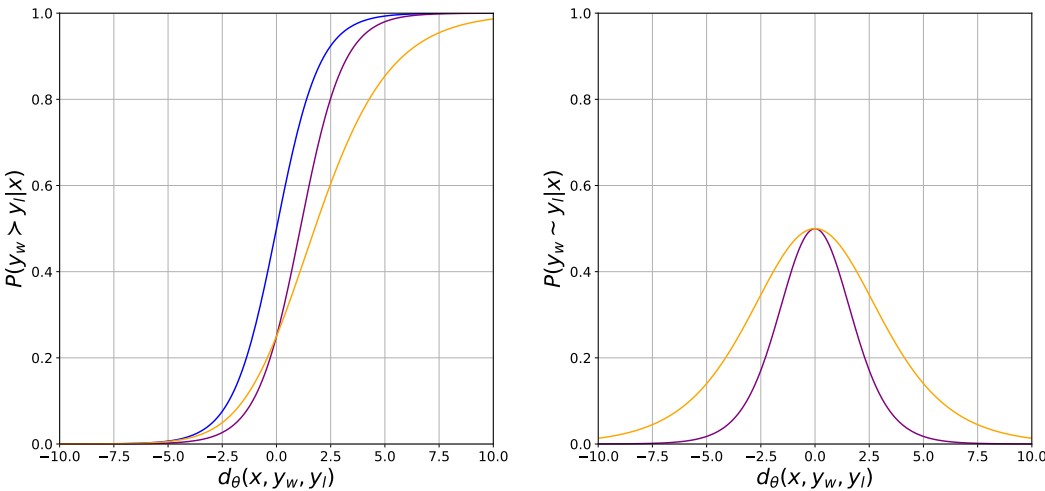

Figure 3: The clear preference probabilities $P(y_w \succ y_l|x)$ (left) and tie probabilities $P(y_w \sim y_l|x)$ (right) as a function of reward margins $d_\theta(x, y_w, y_l)$ for Bradley-Terry (as used in DPO) (blue), Rao-Kupper (purple) (as used in DPO-RK), and Davidson (orange) (as used in DPO-D). $\alpha_{RK} = \log 3$ and $\nu_D = 1$ are used in producing these plots.

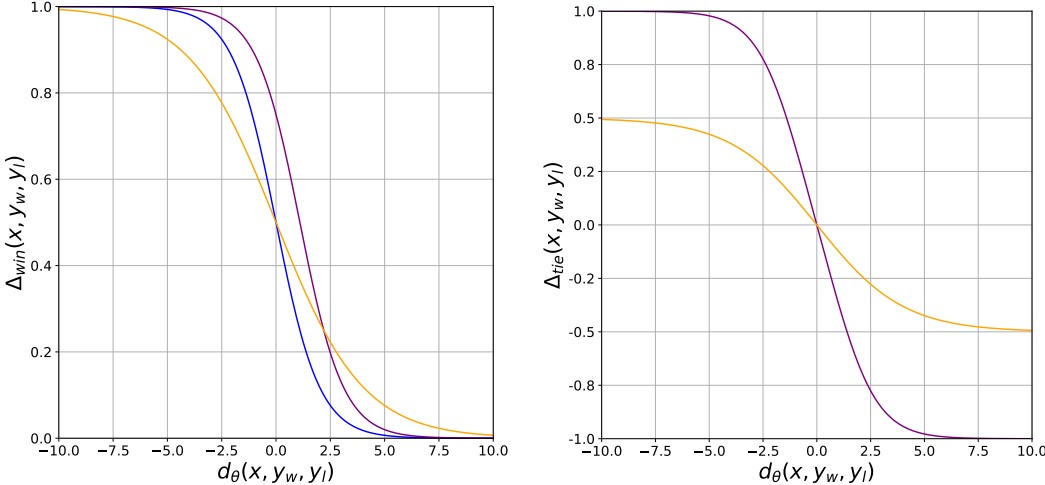

Figure 4: The gradient scale factors for DPO (blue) and DPO-RK (purple) and DPO-D (orange) as a function of reward margins $d_\theta(x, y_w, y_l)$ on clear preference pairs (up) and tied pairs (down). $\alpha_{RK} = \log 3$ and $\nu_D = 1$ are used in producing these plots.

## C  Using DPO-RK and DPO-D model as Preference Classifier

### C.1  Rao-Kupper and Davidson Classifiers

The DPO-RK and DPO-D variants yield probability distributions $p_\theta(y_w \succ_x y_l)$ and $p_\theta(y_w \sim_x y_l)$ in terms of the policy $\pi_\theta$ and the reference model $\pi_{ref}$. We can use these distributions as classifiers to label a pair $(x, y_1, y_2)$ as either a win ($y_1 \succ_x y_2$ or $y_2 \succ_x y_1$) or a tie ($y_1 \sim_x y_2$), whichever has the highest probability under either the Rao-Kupper or the Davidson model (Eqs. 10, 11, or 12, 13). We will evaluate classification performance on held-out data not used in training to see if policies produced by our DPO variants learn to distinguish wins from ties.

## C.2 Preference Pair Classification Accuracy

We assess the performance of the Rao Kupper and Davidson classifiers introduced in Sec.C.1 in terms of their ability to label preference pairs as either clear preferences or ties. Ideally, classification performance will improve: (1) as tied pairs are added to the clear preference data sets (CP vs CP+TP); and (2) with margins generated from models produced by DPO variants that emphasize the distinction between tied pairs and clear preference pairs (DPO-D(CP+TP), DPO-RK(CP+TP)).

We assess classifier performance on the held-out set created by collecting CPs and TPs from the WMT18 ZH-EN test set as was done for WMT20 ZH-EN (Appendix E); this yields pairs with gold labels as either clear preference pairs or tied pairs. Classification and assessment proceeds as follows: we generate reward margins for the WMT18 ZH-EN pairs using DPO(CP), DPO(CP+TP), DPO-RK(CP+TP), DPO-D(CP+TP) models; we use these reward margins to label the unseen pairs using the Davidson and Rao-Kupper classification rules (Sec. C.1); and finally compute the classification accuracy relative to the gold labels.

Results are shown in Table 4. We find that smaller beta in training consistently leads to better overall RK-classification accuracy (+10% overall Acc. from $\beta = 1.0$ to $\beta = 0.1$), suggesting heavy regularization with respect to the reference model impedes preference ranking. Classifiers based on reward margins generated from DPO(CP) models perform well in identifying clear preference pairs (Acc. $> 85\%$) but poorly in identifying tied pairs (Acc. $< 35\%$). This imbalance is likely explained by the DPO(CP) model never having seen tied pairs in training. Adding TPs to the DPO datasets (DPO(CP+TP)) significantly improves the classification accuracy of tied pairs ($+30\%$) with more balanced classification accuracies for CPs and TPs. The best overall classification accuracies ($\approx 73\%$) are obtained with reward margins generated by models trained to match its classifier. Across all beta values, DPO-RK(CP+TP) and DPO-D(CP+TP) achieve better overall accuracy and more-balanced CP accuracy and TP accuracy under their respective decision rules.

| Model | $\beta = 0.1$ | $\beta = 0.5$ | $\beta = 1.0$ |
|---|---|---|---|
| | Rao-Kupper Classifier | | |
| DPO(CP) | 60.1% (**87.1%**, 33.1%) | 52.8% (87.3%, 18.3%) | 50.1% (86.9%, 13.3%) |
| DPO(CP+TP) | 67.0% (72.0%, 62.1%) | 57.5% (69.3%, 45.7%) | 51.5% (71.2%, 31.9%) |
| DPO-RK(CP+TP) | **73.1%** (74.5%, **71.7%**) | 64.2% (73.2%, 55.3%) | 58.5% (73.4%, 43.5%) |
| | Davidson Classifer | | |
| DPO(CP) | 65.3% (**84.4%**, 46.3%) | 57.4% (83.7%, 31.0%) | 53.6% (84.6%, 22.6%) |
| DPO(CP+TP) | 71.0% (59.1%, **82.8%**) | 62.1% (58.3%, 65.8%) | 57.2% (62.3%, 52.2%) |
| DPO-D(CP+TP) | **73.8%** (79.6%, 67.9%) | 66.8% (75.9%, 57.8%) | 62.7% (75.2%, 50.3%) |

Table 4: Preference pair classification accuracies (Overall Acc. (CP Acc., TP Acc.)) for Rao-Kupper and Davidson classification rules based on reward margins computed using DPO(CP), DPO(CP+TP), DPO-RK(CP+TP), and DPO-D(CP+TP) models as evaluated on the WMT18 ZH-EN test set.

## C.3 Empirical Reward Margin Distributions

We now look at the reward margins on held-out pairs to determine how the DPO objective generalizes to unseen data. Ideally, a post-DPO model should assign reward margins that are large for clear preference pairs but close to zero for tied pairs. We assess this on the same held-out data as in the previous section (Sec. C).

In Table 5, reward margins of DPO(CP+TP), DPO-RK(CP+TP), and DPO-D(CP+TP) are similar and well-behaved, showing means close-to-zero on TPs ($< 0.4$) and farther from zero for CPs ($> 2.3$). Reward margin standard deviations are also similar and reasonably small. However the standard deviation for both tied pairs and clear preference pairs are much higher for DPO(CP) models ($\geq 11.1$ on CPs and $\geq 7.9$ on TPs).

This can be explained by Figure 5 which shows that DPO(CP) models overwhelmingly assign preference probability values of either $\sim 1.0$ or $\sim 0.0$ to tied pairs, corresponding to very positive and very negative reward margins, respectively. This contributes to the high standard deviation and shows that for a tied pair $(y_1, y_2)$, DPO(CP) model exhibits a strong preference for either $y_1 \succ y_2$ or

| Model | $\beta = 0.1$ | $\beta = 0.5$ | $\beta = 1.0$ | $\beta = 0.1$ | $\beta = 0.5$ | $\beta = 1.0$ |
|---|---|---|---|---|---|---|
| | Clear Preference Pairs | | | Tied Pairs | | |
| DPO(CP) | 8.2 ±12.0 | 9.5 ±13.2 | 10.0 ±11.1 | 0.7 ±13.2 | 0.6 ±9.4 | 0.4 ±7.9 |
| DPO(CP+TP) | 2.4 ±3.3 | 2.3 ±3.2 | 2.5 ±3.3 | 0.4 ±4.8 | 0.3 ±3.2 | 0.2 ±2.7 |
| DPO-RK(CP+TP) | 2.9 ±4.3 | 2.8 ±3.3 | 3.0 ±3.3 | 0.0 ±1.3 | 0.0 ±1.4 | 0.0 ±1.7 |
| DPO-D(CP+TP) | 4.6 ±5.8 | 4.8 ±6.1 | 4.9 ±6.3 | 0.0 ±2.0 | 0.1 ±2.3 | 0.0 ±2.4 |

Table 5: Reward margin statistics (mean ± std) for held-out Clear Preference Pairs and Tied Pairs collected from WMT18 ZH-EN.

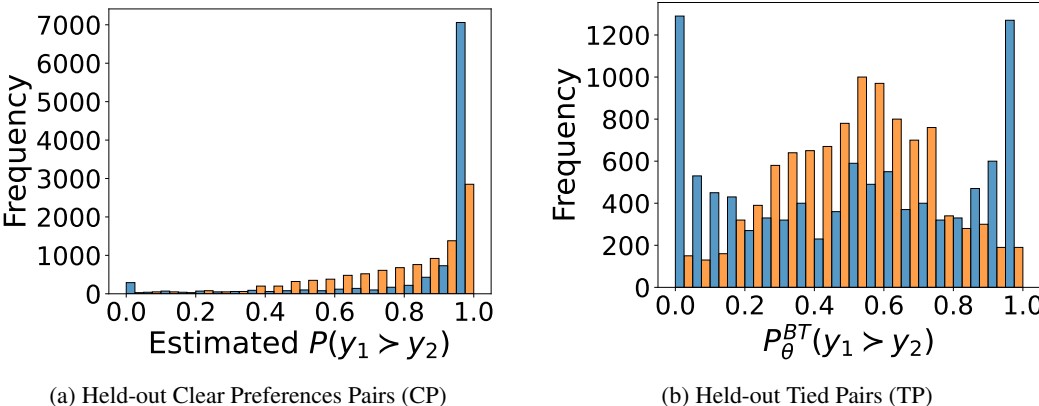

(a) Held-out Clear Preferences Pairs (CP)   (b) Held-out Tied Pairs (TP)

Figure 5: Empirical distribution of preference probabilities under the Bradley-Terry model using the implicit reward function from the trained DPO policy on heldout CPs and TPs. DPO(CP) in blue, and DPO(CP+TP) in orange.

$y_2 \succ y_1$, even though these are tied pairs by construction ($y_1 \sim y_2$). In contrast, DPO(CP+TP) yields well-behaved estimated preference probability distribution more centered around 0.5 for tied pairs.

The DPO(CP) model correctly assigns high preference probability to most of the held-out CPs. This is consistent with its high classification accuracy on clear preference pairs in Table 4. Similar to the estimated preference probability on held-out TPs, the DPO(CP) model tends to give confident, clear preference judgment with $> 0.8$ probability in either direction. In comparison, the DPO(CP+TP) model is more conservative in making preference judgments, showing a less-sharp preference probability distribution over the held-out CP pairs. These results suggest that incorporating ties in DPO training leads to preference probability distributions that more evenly spread on both CPs and TPs as opposed to one concentrated on the two ends.

For completeness, we also show the clear preference/tie probability distributions produced by models trained with DPO-RK(CP+TP) and DPO-D(CP+TP) on held-out clear preference pairs and tied pairs. Figure 6 show that these distributions are well-behaved in that most of the probability mass are allocated to $P_\theta(y_1 \succ y_2) > 0.5$ on held-out clear preference pairs and to $P_\theta(y_1 \sim y_2) \approx 0.5$ on held-out tied pairs. We note that under our hyper-parameter setting for the Rao-Kupper and Davidson models, the maximal tie probability is 0.5.

All models in this analysis are trained with $\beta = 0.1$.

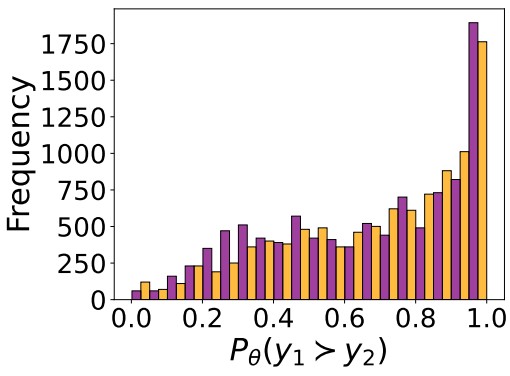 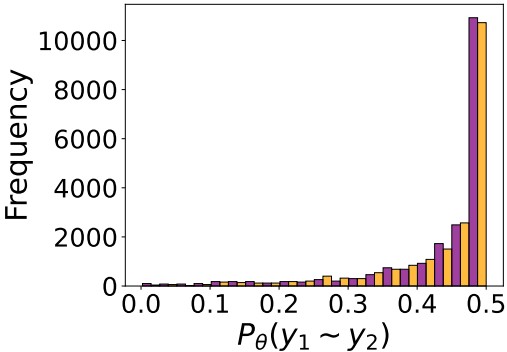

(a) Preference probability under the models on held-out clear preference pairs.

(b) Tie probability under the models on held-out tied pairs.

Figure 6: DPO-D (orange) and DPO-RK (purple) preference/tie probability on held-out sets under the Davidson and Rao-Kupper models, respectively.

## D  Training Dynamics and Convergence Behaviour

We analyse how the inclusion of tied pairs affects the training dynamics of DPO, DPO-RK, and DPO-D. We study the BLOOMZ-mt-7b datasets with $\beta = 0.7$ for WMT21 ZH-EN as these systems show both strong regularization effects and task performance degradation when tied pairs are added. Figure 7 shows the evolution of reward margins and gradient scale factors (Eqs.17 - 21).

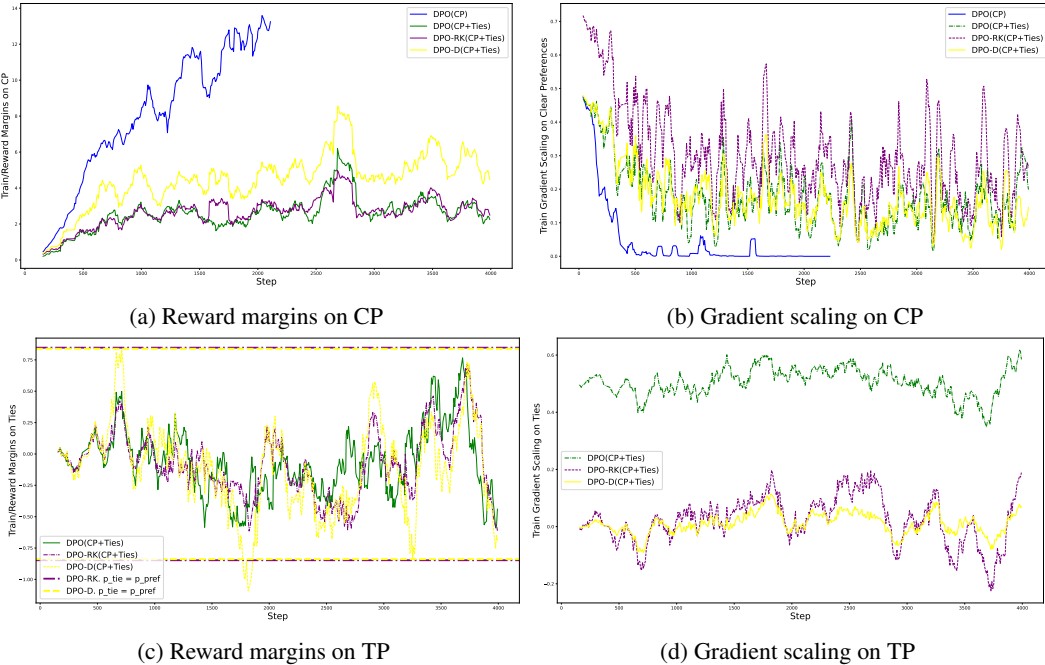

(a) Reward margins on CP

(b) Gradient scaling on CP

(c) Reward margins on TP

(d) Gradient scaling on TP

Figure 7: DPO(CP) (blue), DPO(CP+TP) (green), DPO-RK(CP+TP) (purple), and DPO-D(CP+TP) (yellow) training statistics on WMT21 ZH-EN.

DPO(CP) is well behaved: the reward margins on the CP data increase over the epoch; the DPO losses on the CP dataset decrease over the epoch; and the DPO gradient scale factor shows that learning slows and stabilizes after the $500^{th}$ batch.

Adding tied pairs to the DPO dataset alters this behaviour for both tied pairs and clear preference pairs. DPO(CP+TP) does yield some gains in reward margins for clear preference pairs, but these are

well below that of DPO(CP). By contrast, DPO(CP+TP) fails almost entirely to find any improvement in the reward margins for its tied pair data. While this is less than ideal from a modelling perspective, we note that it provides empirical support for the observation in the previous section that the reward margins on tied pairs should ideally remain close to zero. Similar behaviour is observed in the DPO loss. Decreases in loss over clear preference pairs are offset by loss increases on the tied pairs. This is reflected in the gradient scale factors. The gradient scale factors remain high as DPO(CP+TP) searches for a better policy.

DPO-RK(CP+TP) and DPO-D(CP+TP) exhibit largely the same dynamics as DPO(CP+TP) except that the gradient scale factors on TPs have mean close to 0 instead of 0.5 and fluctuate between positive and negative values. This showcases that DPO-RK and DPO-D work to drive reward margin on TPs close to zero by possibly reversing optimization direction, a mechanism not present in the original DPO.

# E    Experimental Details and Full Results

We provide additional details of our experiments on Neural Machine Translation and Summarization with respect to the SFT models, the training configurations, and the decoding procedures. All experiments are run with the random seed set to 0. Codes are available at `https://github.com/EriChen0615/DPO-RKD`.

## E.1    Neural Machine Translation

We largely follow Yang et al. [2024c] in our experimental setup for NMT where the preference dataset is obtained via sampling and BLEURT-based ranking as explained in Sec.3.1.

**CPs and TPs in NMT.**    We use DPO to improve translation quality similar to that done in Yang et al. [2024c]. We apply DPO with BLOOMZ-mt-7b [Muennighoff et al., 2023] as the baseline model. Translation quality is measured with BLEURT [Sellam et al., 2020] and COMET [Rei et al., 2020] on the WMT21 ZH-EN and IWSLT17 FR-EN translation test sets. We note that the WMT-23 metric overview paper reports high correlations (0.776 and 0.779, Table 1, Freitag et al. [2023]) between BLEURT and COMET and human judgment. To construct a DPO preference dataset for the WMT21 ZH-EN test set, we use BLOOMZ-mt-7b to generate 32 translations (via sampling) for each source sentence in the WMT20 ZH-EN test set. For each source sentence, the translations are ranked by their BLEURT scores computed with respect to the reference translations. The highest and lowest scoring translations form the Clear Preference Pairs; for each source sentence, these are the two translations with the greatest difference in BLEURT score. By contrast, we take the Tied Pairs as the two non-identical translations with the minimum absolute BLEURT difference; the translation with higher BLEURT is labeled as the winner of each Tied Pair. This yields ca. 16K CPs and TPs for use in DPO. The same procedure is applied to the IWSLT17 validation set, yielding ca. 800 CPs and TPs for use as DPO preference datasets. We validate the selected TPs using GPT-4 in Appendix F.8.

**SFT Models**    On WMT-21 ZH-EN, we performed supervised fine-tuning on the BLOOMZ-mt-7b Muennighoff et al. [2023] using previous WMT test sets to obtain the SFT model from which we train with DPO/DPO-RK/DPO-D. The clear preference pairs and tied pairs are generated by sampling from this SFT model. On IWSLT-17 FR-EN, we use the pretrained BLOOMZ-mt-7b model directly in sampling clear preferences and tied pairs and in DPO fine-tuning, as we find further SFT leads to repetitive generation.

**Training Details**    We use the RMSProp optimizer with the learning rate set to $5e^{-7}$ and the number of warm-up steps set to 150. All NMT experiments are run on two Nvidia A100-80G GPUs with an effective batch size of 4. We used FP32 for training the policy. The log-probabilities from the reference model are pre-computed with FP32 precision. Each training run takes $\approx 2$ hours on WMT20 ZH-EN CP+TP data and $\approx 1$ hour on IWSLT17 FR-EN data.

**Decoding**    Following Yang et al. [2024c], we use beam search with a beam size = 4 to decode all models.

**Held-out Clear Preference Pairs and Tied Pairs**    We curate held-out sets by generating translations by sampling on the WMT18 ZH-EN test set. Clear Preference Pairs and Tied Pairs are identified using their rankings under BLEURT exactly as done for WMT21 ZH-EN. This gives 3980 CPs and 3980 TPs for held-out evaluation in Appendix C.

## E.2    Experiments with Stronger Translation Systems

We conducted additional experiments, replacing the BLOOMZ-mt-7B model with the ALMA-7B-LoRA model reported in Xu et al. [2024], which has better translation performance.

We repeated the procedure described in Section 3.1 to collect 3000 Clear Preferences (CPs) and 3000 Tied Pairs (TPs) for ZH-EN translation. This was done by sampling candidate translations using ALMA-7B-LoRA on the ZH-EN training set of Xu et al. [2024], followed by pair selection based on scores from automatic metrics. Following Xu et al. [2024], we used the average score of KIWI-XXL and XCOMET to rank responses. We evaluated our models on the ZH-EN test set of Xu et al. [2024], which consists of 2009 parallel sentences selected from FLORES-200. We report COMET, KIWI-XXL, and XCOMET as evaluation metrics. We also report KL divergence with respect to the reference model.

The table below shows the best-performing system evaluated on ZH-EN translation obtained from a $\beta$ sweep in $[0.1, 0.3, 0.5, 0.7, 0.9]$ under each configuration. For reference, we provide the performance of the base ALMA-7B-LoRA model as reported in Xu et al. [2024]. Additionally, ALMA-7B-R represents the best 7B model reported in Xu et al. [2024], trained with Contrastive Preference Optimization (CPO) on a larger dataset than the one we used. While it is not directly comparable, it serves as a strong translation baseline.

| Model | KL | COMET | KIWI-XXL | XCOMET | Mean(X+K) |
|---|---|---|---|---|---|
| ALMA-7B-LoRA Xu et al. [2024] | - | 79.8 | 73.7 | 83.9 | 78.8 |
| ALMA-7B-R Xu et al. [2024] | - | 81.0 | 75.7 | 90.7 | 83.2 |
| **Our Systems** | | | | | |
| DPO(CP) | 101.64 | 80.5 | 75.2 | 91.1 | 83.2 |
| DPO(CP+TP) | 60.37 | 80.2 | 75.0 | 90.8 | 82.9 |
| DPO-RK(CP+TP) | 75.57 | 80.2 | 75.0 | 90.8 | 82.9 |
| DPO-D(CP+TP) | 30.92 | 80.5 | 75.9 | 90.8 | 83.4 |

Table 6: Performance metrics for ZH-EN translation systems. Results include KL divergence, COMET, KIWI-XXL, and XCOMET scores, along with the mean of XCOMET and KIWI-XXL. ALMA-7B-LoRA and ALMA-7B-R results are from Xu et al. [2024].

We achieve the best performance under the metrics with DPO-D(CP+TP), which also has the lowest KL divergence, on top of a very strong baseline. Consistent with all our other results, including Tied Pairs in all training procedures yields a strong regularization effect.

## E.3    Summarization

**CPs and TPs in Summarization.**    We follow Amini et al. [2024b] in DPO fine-tuning of Pythia-2.8B [Biderman et al., 2023] on the TL;DR dataset [Stiennon et al., 2020] with evaluation via win-rate against human-written summaries. Previous works use GPT-4 to compute the win-rate [Rafailov et al., 2023, Amini et al., 2024a]. We find that the judgments of PairRM [Jiang et al., 2023] agree well with those of GPT-4 (Appendix F.6) and opt to use PairRM win-rate as a cost-effective automatic metric. In the TL;DR task, each prompt is associated with a collection of paired summaries, with a winner and a loser identified for each pair. There is no immediately obvious way to distinguish TPs from CPs in the collection and so we use DPO itself to select TPs. We first apply DPO with $\beta = 0.1$ on the full TL;DR training dataset. Using the reward model formed by this model and the reference model, we compute the reward margins of all pairs of summaries in the training split. For each prompt, the pair with minimal reward margin is treated as a TP, with all other pairs kept as CPs, yielding ca. 14k (15.3%) TPs. See Appendix F.7 for a study of this selection strategy.

We follow Amini et al. [2024b] in experimental setups. The preference dataset is obtained via sampling and ranking with a DPO model without requiring an external reward model as explained in Sec.3.1.

**SFT Model**    We follow Amini et al. [2024b] to supervise-finetune a Pythia-2.8B model Biderman et al. [2023] on the chosen responses in TL;DR train split for one epoch to obtain the initial checkpoint for preference learning. We use the summarization prompt provided in Appendix D.2 by Rafailov et al. [2023].

**Training Details**    We use the RMSProp optimizer with the learning rate set to $5e^{-7}$ and the number of warm-up steps set to 150. All summarization experiments are run on two Nvidia A100-40G GPUs with an effective batch size of 64. We used FP32 for the policy and FP16 for the reference model. Each training run takes $\approx 7$ hours on TL;DR CP+TP data.

**Decoding**    We use greedy decoding for all models as we find it performs on-par or better than temperature sampling (Appendix F.6).

### E.4    Tabulated KL-Performance Results on NMT and Summarization

We tabulate the KL-Performance results shown in Figure 1 and Figure 2.

#### E.4.1    Neural Machine Translation

In addition to KL Divergence and BLEURT, we also provide COMET [Rei et al., 2020] scores, BLEU [Post, 2018] scores and BLEU's Length Ratio.

We observe the "reward hacking" phenomenon identified by Yang et al. [2024c] on both WMT21 ZH-EN and IWSLT17 FR-EN where systems achieve good BLEURT but have large length ratio (>1.5) and lower COMET than the pre-DPO system. These systems learn to generate long, repetitive translations which BLEURT fails to recognize as low-quality. Yang et al. [2024c] find that using small beta values (e.g. 0.1) in DPO training results in reward hacking models. Our results are consistent with their findings and further suggest that large KL divergence from the reference model is a good indicator for reward hacking. On WMT21 ZH-EN, the only model that exhibits reward hacking is trained by DPO(CP) with beta=0.1 which also yields the highest KL divergence (174.13) among all models, greatly exceeding the second-highest KL divergence (68.12). On IWSLT17 FR-EN, Almost all models with KL Divergence > 30 (DPO(CP), $\beta = 0.1$, DPO-RK(CP+TP), $\beta = 0.1$ and DPO-D(CP+TP) $\beta = 0.1, 0.5$) show reward hacking behaviours.

Reward hacking on NMT can be resolved by increasing regularization with respect to the reference model. We find that training with larger beta values or incorporating ties in DPO-RK/DPO-D training can provide such regularization without performance degradation.

#### E.4.2    Summarization

Table 9 shows the KL-PairRM winrate on TL;DR summarization.

| System | beta | KL Divergence | BLEU | Length Ratio | COMET | BLEURT |
|---|---|---|---|---|---|---|
| Bloomz-mt-7b1-SFT | - | 0 | 17.6 | | 77.9 | 61.6 |
| DPO(CP) | 0.1 | 174.13 | 7.23 | 3.01 | 70.2 | 67.7 |
| DPO(CP) | 0.2 | 68.12 | 20.8 | 1.10 | 80.8 | 66.2 |
| DPO(CP) | 0.3 | 62.85 | 20.7 | 1.13 | 80.6 | 66.4 |
| DPO(CP) | 0.4 | 56.02 | 21.4 | 1.09 | 80.7 | 66.4 |
| DPO(CP) | 0.5 | 50.99 | 21.2 | 1.11 | 80.8 | 66.5 |
| DPO(CP) | 0.6 | 47.97 | 21.5 | 1.09 | 80.9 | 66.5 |
| DPO(CP) | 0.7 | 44.08 | 21.5 | 1.11 | 81.0 | 66.7 |
| DPO(CP) | 0.8 | 41.88 | 21.3 | 1.14 | 80.8 | 66.7 |
| DPO(CP) | 0.9 | 41.24 | 21.5 | 1.14 | 80.8 | 66.8 |
| DPO(CP) | 1.9 | 33.69 | 22.3 | 1.09 | 81.2 | 67.0 |
| DPO(CP) | 1.2 | 32.01 | 22.4 | 1.09 | 81.3 | 67.1 |
| DPO(CP) | 1.5 | 29.58 | 21.7 | 1.13 | 81.1 | 67.1 |
| DPO(CP) | 1.55 | 29.01 | 21.9 | 1.13 | 81.1 | 67.1 |
| DPO(CP+TP) | 0.1 | 51.29 | 20.3 | 1.16 | 80.0 | 66.0 |
| DPO(CP+TP) | 0.2 | 36.37 | 18.8 | 1.30 | 80.1 | 66.6 |
| DPO(CP+TP) | 0.3 | 26.15 | 19.5 | 1.24 | 80.2 | 66.6 |
| DPO(CP+TP) | 0.4 | 18.21 | 20.6 | 1.20 | 80.4 | 66.6 |
| DPO(CP+TP) | 0.5 | 15.47 | 21.2 | 1.15 | 80.4 | 66.4 |
| DPO(CP+TP) | 0.6 | 14.74 | 21.9 | 1.10 | 80.6 | 66.4 |
| DPO(CP+TP) | 0.7 | 13.29 | 22.1 | 1.11 | 80.5 | 66.4 |
| DPO(CP+TP) | 0.8 | 12.57 | 22.2 | 1.10 | 80.5 | 66.2 |
| DPO(CP+TP) | 0.9 | 12.10 | 21.9 | 1.10 | 80.5 | 66.3 |
| DPO(CP+TP) | 1.0 | 11.43 | 22.0 | 1.11 | 80.5 | 66.2 |
| DPO-RK(CP+TP) | 0.1 | 48.55 | 19.3 | 1.22 | 80.2 | 66.9 |
| DPO-RK(CP+TP) | 0.2 | 28.61 | 22.1 | 1.11 | 80.9 | 66.9 |
| DPO-RK(CP+TP) | 0.3 | 20.21 | 22.5 | 1.11 | 81.0 | 67.1 |
| DPO-RK(CP+TP) | 0.4 | 14.80 | 22.4 | 1.12 | 81.1 | 67.1 |
| DPO-RK(CP+TP) | 0.5 | 11.66 | 22.8 | 1.10 | 81.0 | 67.1 |
| DPO-RK(CP+TP) | 0.6 | 9.74 | 22.2 | 1.13 | 80.8 | 66.8 |
| DPO-RK(CP+TP) | 0.7 | 8.04 | 22.3 | 1.12 | 80.8 | 66.7 |
| DPO-RK(CP+TP) | 0.8 | 8.10 | 22.1 | 1.13 | 80.8 | 66.8 |
| DPO-RK(CP+TP) | 0.9 | 7.58 | 21.8 | 1.15 | 80.7 | 66.8 |
| DPO-RK(CP+TP) | 1.0 | 6.31 | 22.3 | 1.11 | 80.7 | 66.6 |
| DPO-D(CP+TP) | 0.2 | 42.74 | 21.4 | 1.13 | 80.8 | 66.6 |
| DPO-D(CP+TP) | 0.3 | 38.56 | 21.2 | 1.15 | 80.2 | 66.5 |
| DPO-D(CP+TP) | 0.4 | 17.01 | 22.5 | 1.11 | 81.0 | 67.1 |
| DPO-D(CP+TP) | 0.5 | 20.20 | 22.7 | 1.10 | 81.1 | 67.1 |
| DPO-D(CP+TP) | 0.6 | 26.85 | 22.3 | 1.10 | 81.1 | 66.9 |
| DPO-D(CP+TP) | 0.7 | 14.97 | 22.6 | 1.11 | 81.1 | 67.1 |
| DPO-D(CP+TP) | 0.8 | 13.33 | 22.7 | 1.11 | 81.1 | 67.1 |
| DPO-D(CP+TP) | 1.0 | 10.05 | 22.3 | 1.12 | 80.9 | 67.0 |

Table 7: KL-Performance evaluated on WMT-21 ZH-EN.

| System | beta | KL Divergence | BLEU | Length Ratio | COMET | BLEURT |
|---|---|---|---|---|---|---|
| Bloomz-mt-7b1 | - | | 17.6 | | 85.4 | 74.8 |
| DPO(CP) | 0.1 | 53.60 | 25.8 | 1.40 | 82.3 | 74.7 |
| DPO(CP) | 0.3 | 30.80 | 23.7 | 1.60 | 83.6 | 76.5 |
| DPO(CP) | 0.5 | 16.70 | 36.8 | 1.00 | 86.1 | 76.2 |
| DPO(CP) | 0.7 | 13.80 | 38.5 | 1.00 | 86.4 | 76.4 |
| DPO(CP) | 1.0 | 12.40 | 38.6 | 1.00 | 86.5 | 76.5 |
| DPO(CP) | 1.2 | 11.80 | 38.8 | 0.98 | 86.5 | 76.5 |
| DPO(CP) | 1.5 | 10.70 | 38.9 | 0.99 | 86.5 | 76.5 |
| DPO(CP+TP) | 0.1 | 35.60 | 35.8 | 1.00 | 85.6 | 75.5 |
| DPO(CP+TP) | 0.3 | 25.80 | 35.7 | 1.10 | 85.4 | 75.9 |
| DPO(CP+TP) | 0.5 | 22.00 | 35.1 | 1.10 | 85.8 | 76.3 |
| DPO(CP+TP) | 0.7 | 17.00 | 38.7 | 1.00 | 86.3 | 76.3 |
| DPO(CP+TP) | 1.0 | 11.50 | 38.9 | 1.00 | 86.4 | 76.4 |
| DPO(CP+TP) | 1.2 | 8.50 | 39.1 | 0.98 | 86.5 | 76.4 |
| DPO(CP+TP) | 1.5 | 6.30 | 39.0 | 0.98 | 86.4 | 76.3 |
| DPO-RK(CP+TP) | 0.1 | 46.70 | 23.0 | 1.60 | 78.7 | 76.3 |
| DPO-RK(CP+TP) | 0.2 | 19.51 | 35.9 | 1.05 | 85.9 | 76.4 |
| DPO-RK(CP+TP) | 0.3 | 15.50 | 36.1 | 1.10 | 86.1 | 76.5 |
| DPO-RK(CP+TP) | 0.5 | 13.30 | 31.4 | 1.20 | 85.7 | 76.6 |
| DPO-RK(CP+TP) | 0.7 | 10.90 | 31.3 | 1.20 | 85.8 | 76.5 |
| DPO-RK(CP+TP) | 0.8 | 10.90 | 29.9 | 1.28 | 85.6 | 76.5 |
| DPO-RK(CP+TP) | 0.9 | 11.60 | 27.2 | 1.40 | 85.3 | 76.4 |
| DPO-RK(CP+TP) | 1.0 | 11.60 | 26.1 | 1.50 | 85.1 | 76.3 |
| DPO-RK(CP+TP) | 1.2 | 11.80 | 24.4 | 1.57 | 84.8 | 76.3 |
| DPO-D(CP+TP) | 0.1 | 48.60 | 25.3 | 1.41 | 82.6 | 76.3 |
| DPO-D(CP+TP) | 0.3 | 19.90 | 35.4 | 1.07 | 85.8 | 76.5 |
| DPO-D(CP+TP) | 0.5 | 51.90 | 8.4 | 4.35 | 75.1 | 76.1 |
| DPO-D(CP+TP) | 0.7 | 12.80 | 36.6 | 1.06 | 86.2 | 76.6 |
| DPO-D(CP+TP) | 1.0 | 10.30 | 37.8 | 1.03 | 86.3 | 76.6 |
| DPO-D(CP+TP) | 1.2 | 10.90 | 32.1 | 1.20 | 85.9 | 76.6 |

Table 8: KL-Performance evaluated on IWSLT17 FR-EN

| System | beta | KL Divergence | PairRM Winrate |
|---|---|---|---|
| Pythia-2.8B-SFT, Greedy | - | 0.00 | 37.5 |
| DPO(CP) | 0.025 | 97.03 | 67.9 |
| DPO(CP) | 0.05 | 60.31 | 70.3 |
| DPO(CP) | 0.07 | 57.14 | 71.5 |
| DPO(CP) | 0.08 | 38.16 | 66.4 |
| DPO(CP) | 0.10 | 26.82 | 62.5 |
| DPO(CP) | 0.30 | 9.97 | 63.7 |
| DPO(CP) | 0.50 | 5.79 | 59.0 |
| DPO(CP) | 0.70 | 3.78 | 57.8 |
| DPO(CP+TP) | 0.025 | 87.66 | 63.7 |
| DPO(CP+TP) | 0.03 | 119.60 | 66.8 |
| DPO(CP+TP) | 0.04 | 70.69 | 69.5 |
| DPO(CP+TP) | 0.05 | 35.39 | 63.3 |
| DPO(CP+TP) | 0.10 | 17.21 | 57.4 |
| DPO(CP+TP) | 0.30 | 4.50 | 58.6 |
| DPO(CP+TP) | 0.50 | 7.61 | 57.8 |
| DPO(CP+TP) | 0.70 | 2.91 | 55.9 |
| DPO-RK(CP+TP) | 0.04 | 80.86 | 65.2 |
| DPO-RK(CP+TP) | 0.05 | 62.57 | 67.2 |
| DPO-RK(CP+TP) | 0.10 | 40.50 | 67.6 |
| DPO-RK(CP+TP) | 0.20 | 22.24 | 67.6 |
| DPO-RK(CP+TP) | 0.30 | 12.45 | 68.0 |
| DPO-RK(CP+TP) | 0.50 | 6.15 | 65.6 |
| DPO-RK(CP+TP) | 0.70 | 4.33 | 61.7 |
| DPO-D(CP+TP) | 0.05 | 82.35 | 64.8 |
| DPO-D(CP+TP) | 0.10 | 54.06 | 71.5 |
| DPO-D(CP+TP) | 0.20 | 39.23 | 66.0 |
| DPO-D(CP+TP) | 0.30 | 22.46 | 68.8 |
| DPO-D(CP+TP) | 0.40 | 12.57 | 67.6 |
| DPO-D(CP+TP) | 0.50 | 9.92 | 67.2 |
| DPO-D(CP+TP) | 0.70 | 6.82 | 64.8 |

Table 9: KL-PairRM winrate against 256 human-written summaries on TL;DR summarization

### E.5 Standard Deviation on GSM8K Mathematical Reasoning

| $\beta$ | DPO (CP) | DPO-RK (CP+TP) | DPO-D (CP+TP) |
|---|---|---|---|
| 0.1 | $80.4\% \pm 1.34\%$ | $82.9\% \pm 0.90\%$ | $82.2\% \pm 1.95\%$ |
| 0.3 | $83.7\% \pm 0.45\%$ | $84.3\% \pm 0.18\%$ | $83.7\% \pm 0.68\%$ |
| 0.5 | $83.5\% \pm 0.14\%$ | $84.1\% \pm 0.08\%$ | $84.2\% \pm 0.15\%$ |
| 0.7 | $83.6\% \pm 0.53\%$ | $83.4\% \pm 0.19\%$ | $84.5\% \pm 0.58\%$ |
| 1.0 | $83.8\% \pm 0.29\%$ | $84.0\% \pm 0.08\%$ | $83.7\% \pm 0.42\%$ |

Table 10: Median accuracy ($\pm$ standard deviation) on GSM8K test set across five random seeds for each $\beta$ value. Tie-compatible variants (DPO-RK, DPO-D) consistently outperform the baseline DPO.

We run experiments for five random seeds for each variant at each $\beta$ value (75 training runs in total). In Table 10, we report the median accuracy and standard deviation. Consistent with our previous finding, tie-compatible variants achieve higher median accuracy at all $\beta$ values. Gains are generally greater than one standard deviation.

## F Additional Supporting Experiments and Analyses

### F.1 Sensitivity Study on the Value of $\alpha$ for DPO-RK and DPO-D

In our early experiments with a weaker translation model, we find that the choice of $\alpha$ does not have a strong impact on final task performance. In Table 11, we vary the hyperparameter $\alpha$ and report the KL divergence, KIWI-22, BLEURT, and COMET scores with the BLOOMZ-mt-7B setup evaluated on WMT-21 ZH-EN.

| $\alpha$ | KL | KIWI-22 | BLEURT | COMET |
|---|---|---|---|---|
| **DPO-RK** | | | | |
| 0.5 | 11.23 | 78.9 | 66.9 | 81.0 |
| $\ln 3$ | 14.80 | 79.0 | 67.1 | 81.1 |
| 2.0 | 21.84 | 79.1 | 67.1 | 81.1 |
| 3.0 | 30.41 | 79.0 | 66.9 | 80.9 |
| 5.0 | 76.66 | 69.9 | 68.9 | 74.6 |
| **DPO-D** | | | | |
| -2.5 | 11.56 | 79.0 | 67.0 | 81.0 |
| -1.0 | 15.25 | 79.0 | 67.1 | 81.1 |
| 0.0 | 11.66 | 79.2 | 67.1 | 81.0 |
| 1.0 | 29.72 | 79.0 | 67.0 | 81.0 |
| 2.5 | 47.20 | 79.2 | 67.1 | 81.1 |

Table 11: Effect of varying $\alpha$ on DPO-RK ($\beta = 0.4$) and DPO-D ($\beta = 0.5$) systems. For DPO-RK, $\alpha = \alpha_{RK}$ in Eq.15; For DPO-D, $\alpha = \ln \nu_D$ in Eq.16. Results are reported for KL divergence, KIWI-22, BLEURT, and COMET metrics. Default values of $\alpha$ are underlined.

We find that (1) The final task performance is stable over a range of $\alpha$ values (note that $\alpha < 0$ is illegal for DPO-RK) and (2) for large values of $\alpha$, there is relatively little regularization as measured by KL divergence.

As explained in Sec.2.3.1, $\alpha$ controls the width of the band in reward margins where there's little or no gradient contributions from the tied pairs whose difference in reward falls within the band. This explains the lack of regularization under large $\alpha$. Given large $\alpha$, the gradient on TPs approach zero regardless of the reward margin of the pair under the current model. As we show in the paper, tied pairs contribute to a regularization effect. This explains why large and hence little gradient contribution from TPs leads to relatively little regularization.

### F.2 Grid search of $\beta$ and $\alpha$ on WMT

We further conduct a grid search over $\beta$ and $\alpha$ values on WMT to investigate their effects. The results are shown in Table 12.

| Variant | $\beta$ | $\alpha$ | KL | COMET |
|---|---|---|---|---|
| DPO-RK (CP+TP) | 0.2 | 0.1 | 19.73 | 80.9 |
|  |  | 0.5 | 22.31 | 81.0 |
|  |  | $\ln 3$ | 28.61 | 80.9 |
|  |  | 2.0 | 37.05 | 81.1 |
|  |  | 3.0 | 49.00 | 81.0 |
| DPO-RK (CP+TP) | 0.4 | 0.1 | 9.54 | 80.9 |
|  |  | 0.5 | 14.80 | 81.0 |
|  |  | $\ln 3$ | 21.84 | 81.1 |
|  |  | 2.0 | 30.41 | 81.1 |
|  |  | 3.0 | 76.66 | 74.6 |
| DPO-RK (CP+TP) | 0.8 | 0.1 | 6.32 | 80.7 |
|  |  | 0.5 | 5.91 | 80.6 |
|  |  | $\ln 3$ | 8.10 | 80.8 |
|  |  | 2.0 | 10.37 | 80.9 |
|  |  | 3.0 | 14.53 | 80.9 |
| DPO-D (CP+TP) | 0.2 | $-2.5$ | 27.46 | 80.9 |
|  |  | $-1.0$ | 34.80 | 80.6 |
|  |  | 0 | 42.74 | 80.8 |
|  |  | 1.0 | 56.83 | 80.6 |
| DPO-D (CP+TP) | 0.4 | $-2.5$ | 14.56 | 81.0 |
|  |  | $-1.0$ | 18.71 | 81.0 |
|  |  | 0 | 17.01 | 81.0 |
|  |  | 1.0 | 33.15 | 81.0 |
|  |  | 2.5 | 56.18 | 80.7 |
| DPO-D (CP+TP) | 0.5 | $-2.5$ | 11.56 | 81.0 |
|  |  | $-1.0$ | 15.25 | 81.1 |
|  |  | 0 | 11.66 | 81.0 |
|  |  | 1.0 | 29.72 | 81.0 |
|  |  | 2.5 | 47.20 | 81.1 |
| DPO-D (CP+TP) | 0.8 | $-2.5$ | 8.16 | 80.7 |
|  |  | $-1.0$ | 9.10 | 80.7 |
|  |  | 0 | 13.33 | 81.1 |
|  |  | 1.0 | 19.54 | 81.0 |
|  |  | 2.5 | 30.48 | 80.8 |

Table 12: Grid search results for $\beta$ and $\alpha$ on WMT showing the corresponding KL divergence and COMET scores.

Consistent with our findings in Appendix F.1, we find that task performance is stable over a range of $\beta$ and $\alpha$ values. These results add further empirical evidence supporting the role of $\beta$ and $\alpha$ as knobs for regularization strength. Controlled for a fixed value of $\beta$, increasing $\alpha$ and $\beta$ generally leads to higher KL divergence with respect to the reference model at the end of training.

Based on these experiments, we recommend the following hyperparameter tuning scheme for DPO-RK and DPO-D:

1. Run training across a range of $\beta$ values with the default $\alpha$ to identify the effective range of $\beta$.

2. Adjust $\alpha$ (and $\beta$, if desired) at those values, using the final KL divergence as a guide.

As shown in experiments, there is typically a "sweet range" of KL where performance is optimal. This procedure allows practitioners to efficiently identify the best performance–KL trade-off for a given setup.

### F.3 GPT-4 Evaluation on Summarization

For TL;DR, we additionally report the win-rate as evaluated by GPT-4 of the best-performing systems in Table 13.

| Model | PairRM | GPT-4 |
|---|---|---|
| Pythia-2.8-DPO(CP) | **71.5%** | 62.1% |
| Pythia-2.8-DPO-D(CP+TP) | **71.5%** | **64.1%** |
| Pythia-2.8-DPO(CP+TP) | 69.5% | 62.1% |
| Pythia-2.8-DPO-RK(CP+TP) | 68.0% | 57.8% |
| Pythia-2.8-SFT | 37.5% | 36.7% |

Table 13: Win-rate comparison of the best-performing systems evaluated using PairRM and GPT-4.

We note that the rank based on PairRM win-rate agrees with the rank produced by GPT-4 except that GPT-4 prefers Pythia-2.8-DPO-D(CP+TP).

### F.4 Varying the Percentage of Ties

In Table 14 we vary the proportion of TPs and evaluate on the WMT-21 ZH-EN dataset, keeping the CPs unchanged. We train systems with three $\beta$ values [0.2, 0.4, 0.6] and report the best-performing system in terms of BLEURT. We observe that, under the same $\beta$ value, including more TPs reduces the KL divergence with respect to the reference model at the end of training while maintaining the same performance. This provides further empirical evidence for our analysis in Sec.3.3, where ties regularize training.

| TP% | KL | KIWI-22 | BLEURT | COMET |
|---|---|---|---|---|
| DPO-RK (CP+TP) | | | | |
| 25% | 10.16 | 78.4 | 66.5 | 80.7 |
| 50% | 6.05 | 78.2 | 66.2 | 80.4 |
| 75% | 4.40 | 78.5 | 66.4 | 80.7 |
| DPO-D (CP+TP) | | | | |
| 25% | 17.26 | 78.5 | 66.4 | 80.8 |
| 50% | 10.40 | 78.5 | 66.3 | 80.7 |
| 75% | 6.63 | 78.5 | 66.5 | 80.8 |

Table 14: Regularization effects of varying TPs in DPO-RK and DPO-D systems. KIWI-22 Rei et al. [2022] is a reference-less metric. A TP% of 50% means that we keep 50% of all the TPs, resulting in a CPs:TPs ratio of 2:1.

### F.5 IPO Baselines and KIWI-22 Evaluation on WMT

We additionally evaluate our systems with KIWI-22 [Rei et al., 2022], a popular reference-less neural metric used by recent works in evaluating state-of-the-art (SoTA) machine translation systems [Xu et al., 2024].

| Model | Best BLEURT | Best COMET | Best KIWI-22 |
|---|---|---|---|
| Bloomz-mt-7b1-SFT | 61.6 | 77.9 | 77.3 |
| Bloomz-mt-7b1-DPO (CP) | **67.1** | **81.3** | **82.0** |
| Bloomz-mt-7b1-DPO (CP+TP) | 66.6 | 80.6 | 78.5 |
| Bloomz-mt-7b1-DPO-RK (CP+TP) | **67.1** | 81.0 | 79.0 |
| Bloomz-mt-7b1-DPO-D (CP+TP) | **67.1** | 81.0 | 79.2 |
| Bloomz-mt-7b1-IPO (CP) | 66.6 | 80.6 | 78.6 |
| Bloomz-mt-7b1-IPO (CP+TP) | 66.3 | 80.5 | 78.4 |

Table 15: Evaluation results under BLEURT, COMET, and KIWI-22 metrics for various models. DPO and its variants show clear gains over the SFT baseline across all three metrics.

We perform additional NMT experiments with IPO fine-tuning on CPs (IPO(CP+TP) and CPs+TPs (IPO(CP+TP)). We conduct IPO fine-tuning (using code from https://github.com/eric-mitchell/direct-

preference-optimization) with the a range of beta values [0.025, 0.05, 0.1, 0.3, 0.5, 0.7, 1.0] and report the best performing system as evaluated on the WMT-21 ZH-EN dataset in Table 15. We use identical training hyper-parameters as used in DPO, DPO-RK and DPO-D fine-tuning. We note that these metrics are strongly rated in the WMT-23 metric overview paper [Freitag et al., 2023].

We find that (1) both IPO(CP) and IPO(CP+TP) yield substantial gains compared to the baseline SFT model (+4.7 BLEURT. (2) Adding TPs to IPO causes a small degradation (<-0.3) across all metrics. (3) IPO(CP) performs on par with DPO(CP+TP) but under-performs DPO(CP), DPO-RK(CP+TP), and DPO-D(CP+TP) on all metrics.

## F.6  PairRM as a Proxy Evaluator for GPT-4

| System | GPT-4 | PairRM |
|---|---|---|
| DPO | | |
| T=1.0 | 23.4% | 27.3% |
| T=0.75 | 40.2% | 40.6% |
| T=0.5 | 52.3% | 54.7% |
| T=0.25 | 46.9% | 51.6% |
| T=0.0 | 50.4% | 55.5% |
| | | |
| SFT | | |
| T=1.0 | 22.3% | 23.0% |
| T=0.5 | 37.5% | 38.7% |
| T=0.0 | 36.7% | 39.8% |

Table 16: Win-rate of Pythia-2.8B model SFT/DPO on TL;DR train against 256 human-written summaries as judged by GPT4-0613 and PairRM.

PairRM [Jiang et al., 2023] is a strong reward model that has been shown to be effective in curating preference datasets for iterative DPO training [Tran et al., 2023]. In our experiments on TL;DR summarization, we use the PairRM reward model instead of GPT-4 for comparing generated summaries against human references. In this appendix, we show that win-rate as judged by PairRM is a good proxy for GPT4-0613 [OpenAI et al., 2024] win-rate on the TL;DR dataset Stiennon et al. [2020].

We generate summaries from SFT pythia-2.8B model by sampling at temperature $T = [0.0, 0.5, 1.0]$ and the DPO model ($\beta = 0.1$) trained on TL;DR's full training set at temperature $T = [0.0, 0.25, 0.5, 0.75, 1.0]$. Their win-rates against the 256 human-written summaries in the TL;DR valid-2 split as judged by GPT-4 and PairRM are tabulated in Table 16. We find that the win-rates by GPT-4 and PairRM are similar and that system rankings are generally preserved. We opt to use PairRM as our evaluation metric which enables us to conduct experiments faster and at lower costs.

## F.7  Verifying a Tied Pair Selection Strategy for TL;DR

As explained in Sec. 3.1, we use the reward model associated with the DPO model trained on TL;DR to identify summarizations that are similar in quality. Note that we are performing unsupervised labelling of ties in the DPO training data, which is somewhat more forgiving than the classification task discussed in other sections which requires labelling ties in held-out data not seen in training. We do however assume that the reward model should perform well on the data it was trained on.

To investigate these assumptions, we swap the preferred and the dispreferred responses in all tied pairs to form "reversed Tied Pairs" (rTP). If the responses in TP are truly similar in quality (i.e., it is acceptable to reverse the preference direction), training with DPO(CP+TP) and DPO(CP+rTP) should yield similar performing models. Furthermore, the DPO-RK and DPO-D learning procedures which explicitly model tied pairs should yield better performing model. We conduct experiments on TL;DR. Table 17 shows that the performance relation DPO-D(CP+TP) $\sim$ DPO-RK(CP+TP) $\succ$ DPO(CP+TP) $\sim$ DPO(CP+rTP) indeed holds for TL;DR, which suggests that our Tied Pair selection strategy is reasonable.

| System | PairRM |
| --- | --- |
| DPO(CP+ TP) | 58.6% |
| DPO(CP+rTP) | 60.9% |
| DPO-RK(CP+TP) | 68.0% |
| DPO-D(CP+TP) | 68.8% |

Table 17: Win-rates of Pythia-2.8B model DPO on TL;DR train against 256 human-written summaries as judged by PairRM. Systems were trained on CP+TP or CP+rTP data with DPO, DPO-RK, or DPO-D at fixed $\beta = 0.3$. For DPO-RK and DPO-D learning, rTP is equivalent to TP as there is no preference direction for ties.

### F.8   Using GPT-4 to validate ties on NMT

We conduct the following experiments with GPT-4 to evaluate whether the tied pairs selected by BLEURT are reliable.

We randomly select 128 source sentences on our training set sampled from WMT20 ZH-EN and collect their corresponding CPs and TPs as determined by BLERUT score's ranking. Then, for each source sentence, we present the two pairs of translations to GPT-4o-2024-08-06 and prompt it to decide which pair of translations is a tied pair.

We find that on the 128 randomly selected examples from WMT20 ZH-EN, the GPT-4o model agrees 82.03% of the time with BLEURT's assignment of CP and TP.

An example of our prompt and GPT-4's response is below. In this case, GPT-4 chooses the same pair (Pair A) as TP as BLEURT.

The prompt for GPT-4 and example responses are provided below.

```
Pair A:
   Better Translation: Beijing Youth Daily reporters learned that there were many
       important amendments in the first published " drafts".
   Worse Translation: Beijing Youth Daily reporters learned that the "First Draft" has
       many important modifications.
Pair B:
   Better Translation: The Beijing Youth Daily reporter learned that there were many
       important changes in the Draft for the first time published.
   Worse Translation: Beijing Youth Daily reporter learned that the first version of
       the "Comments" had been revised many times.

You should provide a short rationale and then decide which pair is a tied pair. You
    should follow the format "DECISION: [[A]]" or "DECISION: [[B]]" to give your
    decision.

Response:

To determine which pair represents a tied pair, let's evaluate the differences between
    the translations in each pair.

In Pair A:
- The "better" translation is: "Beijing Youth Daily reporters learned that there were
    many important amendments in the first published 'drafts'."
- The "worse" translation is: "Beijing Youth Daily reporters learned that the 'First
    Draft' has many important modifications."

Both translations in Pair A convey similar meanings, describing modifications or
    amendments to a draft. The primary differences are stylistic, such as the
    placement and quotation of "draft".

In Pair B:
- The "better" translation is: "The Beijing Youth Daily reporter learned that there
    were many important changes in the Draft for the first time published."
- The "worse" translation is: "Beijing Youth Daily reporter learned that the first
    version of the 'Comments' had been revised many times."
```

```
In Pair B, the two translations have significant differences in meaning. The "better"
    translation accurately captures the notion of "important changes" and "first
    published," whereas the "worse" translation introduces "first version of the
    'Comments'" and a focus on revisions rather than initial publication, which
    diverges more from the source sentence.

With this analysis, Pair A has translations that are more similar in meaning and are
    thus likely to be the tied pair, while Pair B shows a clearer distinction in
    translation quality.

DECISION: [[A]]
```
Listing 1: Prompt and Response for using GPT-4 to validate tie selection on NMT

# G   Simulating the Perfect DPO-Davidson Policy

In Section 3.2 we make use of the relationship derived by Chen et al. [2024, Appendix A.2] which specifies the optimal DPO policy to minimize the binary classification loss

$$\min_{\pi} \mathbb{P}(y_1 \succ_x y_2) \log \pi(y_1 \succ_x y_2)]$$
$$+ (1 - \mathbb{P}(y_1 \succ_x y_2)) \log(1 - \pi(y_1 \succ_x y_2))$$

where $\mathbb{P}(y_1 \succ_x y_2)$ is the human ground truth preference distribution.

We extend the analysis of Chen et al. [2024] to include the Davidson model, noting that the binary maximum likelihood objective becomes ternary. We assume we have the ground-truth human preference distributions $\mathbb{P}(y_1 \succ_x y_2)$, $\mathbb{P}(y_2 \succ_x y_1)$, and $\mathbb{P}(y_1 \sim_x y_2)$ needed to define the objective. The resulting Theorem 1 can be viewed as a generalization of Theorem 3 of Chen et al. [2024] that allows for the observations of ties. Where ties are not allowed (i.e. $\nu_D = 0$), the Davidson model simplifies to the Bradley-Terry model and Theorem 3 of Chen et al. [2024] is recovered as a special case of Theorem 1.

**Theorem 1** (Simulating Perfect DPO-D Policy). *Assume we are given an aggregated comparison datapoint $(x, y_1, y_2)$ and human ground-truth preference probabilities $\mathbb{P}(y_1 \succ_x y_2)$, $\mathbb{P}(y_1 \succ_x y_2)$, and $\mathbb{P}(y_1 \sim_x y_2)$ which obey the Davidson model with hyper-parameter $\nu_D$. Let the reference model be $\pi_{ref}$. It follows that the perfect DPO-Davidson policy $\pi^*$ on this aggregated comparison datapoint satisfies*

$$\frac{\pi^*(y_1|x)}{\pi^*(y_2|x)} = \frac{\pi_{ref}(y_1|x)}{\pi_{ref}(y_2|x)} \left( \frac{\mathbb{P}(y_1 \succ_x y_2)}{\mathbb{P}(y_2 \succ_x y_1)} \right)^{1/\beta} \tag{25}$$

*or equivalently*

$$\frac{\pi^*(y_1|x)}{\pi^*(y_2|x)} = \frac{\pi_{ref}(y_1|x)}{\pi_{ref}(y_2|x)} \left( 2\nu_D \frac{\mathbb{P}(y_1 \succ_x y_2)}{\mathbb{P}(y_1 \sim_x y_2)} \right)^{2/\beta} \tag{26}$$

*Proof.* The DPO-D policy objective optimizes the following three-way classification loss:

$$\min_{\pi} \mathbb{P}(y_1 \succ_x y_2) \log \pi(y_1 \succ_x y_2)$$
$$+ \mathbb{P}(y_2 \succ_x y_1) \log \pi(y_2 \succ_x y_1)$$
$$+ \mathbb{P}(y_1 \sim_x y_2) \log \pi(y_1 \sim_x y_2)$$

Let $\theta^*$ denotes a set of parameters such that $\pi_{\theta^*}$ is an optimal policy for the above loss, then $\pi_{\theta^*}$ satisfies:

$$\pi_{\theta^*}(y_1 \succ_x y_2) = \mathbb{P}(y_1 \succ_x y_2)$$
$$\pi_{\theta^*}(y_2 \succ_x y_1) = \mathbb{P}(y_2 \succ_x y_1)$$
$$\pi_{\theta^*}(y_1 \sim_x y_2) = \mathbb{P}(y_1 \sim_x y_2)$$

Expressing the policy probability $\pi_{\theta^*}(y_w \succ_x y_l)$ and $\pi_{\theta^*}(y_l \succ_x y_w)$ in terms of the reward margins $d_{\theta^*}(x, y_w, y_l)$:

$$\mathbb{P}(y_1 \succ_x y_2) = \frac{1}{1 + e^{-d_{\theta^*}(x, y_w, y_l)} + 2\nu_D e^{-d_{\theta^*}(x, y_w, y_l)/2}}$$

$$\mathbb{P}(y_2 \succ_x y_1) = \frac{e^{-d_{\theta^*}(x, y_1, y_2)}}{1 + e^{-d_{\theta^*}(x, y_1, y_2)} + 2\nu_D e^{-d_{\theta^*}(x, y_1, y_2)/2}}$$

Rearranging, we have

$$\frac{\mathbb{P}(y_2 \succ_x y_1)}{\mathbb{P}(y_1 \succ_x y_2)} = \exp\left( -d_{\theta^*}(x, y_1, y_2) \right)$$

$$= \exp\left( \beta \log \frac{\pi_{\theta^*}(y_2|x)}{\pi_{ref}(y_2|x)} - \beta \log \frac{\pi_{\theta^*}(y_1|x)}{\pi_{ref}(y_1|x)} \right)$$

Taking logarithms on both side and divide by $\beta$.

$$\frac{1}{\beta}\log\frac{\mathbb{P}(y_1 \succ_x y_2)}{\mathbb{P}(y_2 \succ_x y_1)} = \log\frac{\pi_{\theta^*}(y_2|x)\pi_{ref}(y_1|x)}{\pi_{ref}(y_2|x)\pi_{\theta^*}(y_1|x)}$$

Exponentiating both sides gives

$$\frac{\pi_{\theta^*}(y_2|x)}{\pi_{\theta^*}(y_1|x)} = \frac{\pi_{ref}(y_2|x)}{\pi_{ref}(y_1|x)}\left(\frac{\mathbb{P}(y_2 \succ_x y_1)}{\mathbb{P}(y_1 \succ_x y_2)}\right)^{1/\beta}$$

Taking the inverse yields Eq 25.

To see the equivalence between Eq 25 and Eq 26, note that the ground-truth preference and tie probabilities which obey the Davidson model satisfy the following relation:

$$\mathbb{P}(y_1 \sim_x y_2) = 2\nu_D\sqrt{\mathbb{P}(y_1 \succ_x y_2)\mathbb{P}(y_2 \succ_x y_1)}$$

Rearranging Eq 25:

$$\begin{aligned}
\frac{\pi^*(y_1|x)}{\pi^*(y_2|x)} &= \frac{\pi_{ref}(y_1|x)}{\pi_{ref}(y_2|x)}\left(\frac{\mathbb{P}(y_1 \succ_x y_2)}{\mathbb{P}(y_2 \succ_x y_1)}\right)^{1/\beta} \\
&= \frac{\pi_{ref}(y_1|x)}{\pi_{ref}(y_2|x)}\left(\sqrt{\frac{\mathbb{P}(y_1 \succ_x y_2)}{\mathbb{P}(y_2 \succ_x y_1)}}\right)^{2/\beta} \\
&= \frac{\pi_{ref}(y_1|x)}{\pi_{ref}(y_2|x)}\left(\frac{\mathbb{P}(y_1 \succ_x y_2)}{\sqrt{\mathbb{P}(y_1 \succ_x y_2)\mathbb{P}(y_2 \succ_x y_1)}}\right)^{2/\beta} \\
&= \frac{\pi_{ref}(y_1|x)}{\pi_{ref}(y_2|x)}\left(2\nu_D\frac{\mathbb{P}(y_1 \succ_x y_2)}{\mathbb{P}(y_1 \sim_x y_2)}\right)^{2/\beta}
\end{aligned}$$

which is Eq 26. $\qquad\square$

# H   Extended Related Work

**Variants of Direct Preference Optimization**   A range of variants of Direct Preference Optimization have been proposed based on problem-specific or theoretical motivations. Park et al. [2024] tackle excessive response length by introducing explicit length normalization in the DPO objective. SimPO [Meng et al., 2024] modifies the DPO objective to remove the need for a reference model and to include length normalization. KTO [Ethayarajh et al., 2024] is motivated by Kahneman and Tversky's prospect theory and learns from non-paired preference data. ODPO [Amini et al., 2024b] incorporates preference strength in the objective by introducing an offset parameter. In deriving ODPO, the offset parameter of Amini et al. [2024a, Theorem 3]) plays a role similar to the sensitivity threshold of Rao and Kupper [1967]. We note that the ODPO objective with a fixed offset agrees with our proposed DPO-RK objective restricted to clear preference data, but does not extend to ties.

**Frameworks for Pair-wise Preference Optimization**   Several works propose theoretical frameworks for understanding general Preference Optimization from which DPO can be obtained as a special case. Azar et al. [2024] introduces the $\Psi$PO formalism which allows alternative expression of the reward in terms of the model's predicted probability. IPO is derived when the identity mapping is used, and DPO arises under a log-ratio mapping. Dumoulin et al. [2024] formulate learning from pair-wise preference as learning the implicit preference generating distribution of the annotators. In this formalism, DPO is a well-specified model for the implicit preference distribution assuming that the human preference generative process follows the Bradley-Terry model. Our work can be viewed as assuming an annotator preference generating distribution that allows for the outcome of a tie (i.e. the Rao-Kupper or the Davidson model). Tang et al. [2024] propose a generalized approach to deriving offline preference optimization losses through binary classification. In this work, we consider the ternary classification with the possibility of declaring a tie. In Appendix G, we show that the 'perfect' DPO-D policy can be simulated starting from the ternary classification loss.

**Pair-wise Comparison Models** Hamilton et al. [2023] review the Bradley-Terry model, including its relation to the logistic distribution [Bradley and Gart, 1962] and Luce choice axiom Luce [1959b]. The Rao-Kupper [Rao and Kupper, 1967] and the Davidson model [David, 1988] are two notable extensions to Bradley-Terry (Sec. 2.2). See the review by David [1988] and bibliography by Davidson and Farquhar [1976]. Modeling ties remains an active research topic in fields such as sport team ranking [Zhou et al., 2022], medical treatments [Gaohong Dong and Vandemeulebroecke, 2020], and chatbots [Ameli et al., 2025].

# I Qualitative Cases from DPO, DPO-RK, and DPO-D Systems

We collect the responses on WMT-21 ZH-EN from the best performing model under each configuration. Below, we present two cases where systems trained with DPO-RK(CP+TP) and DPO-D(CP+TP) perform better than DPO(CP) and one case where they perform worse than DPO(CP) as judged by KIWI-22.

The KIWI-22 score of each translation is shown in parentheses.

**Case 1: ID = 10**

**Source:** 小康不小康，关键看老乡。

**Reference:** Hometowns are the key to moderate prosperity.

- **DPO(CP):** It is not important whether you are rich or poor, as long as you have a good relationship with your neighbors. (71.6)
- **DPO(CP+TP):** Wealthy or poor, it all depends on our neighbors. (72.0)
- **DPO-RK(CP+TP):** Wealthy or poor, it all depends on our neighbors. (72.0)
- **DPO-D(CP+TP):** Whether you are rich or poor depends on your neighbors. (77.9)

In this case, the DPO(CP) system produces a long translation with incorrect meaning. The other three systems produce correct, literal translations. The translation from DPO-D(CP+TP) is more fluent. However, all systems fail to fully capture the meaning of "小康" (i.e., "moderate prosperity") as done in the reference.

**Case 2: ID = 269**

**Source:** 这一航空运营商在声明中说，旗下一架客机7月15日从华盛顿州西雅图飞往得克萨斯州奥斯汀途中，一台发动机突然熄火，客机最终在奥斯汀机场成功紧急降落。

**Reference:** The airline released a statement claiming that as its aircraft was flying from Seattle, Washington to Austin, Texas, one of the engines suddenly stalled, and the aircraft finally made an emergency landing in Austin.

- **DPO(CP):** This airline operator said in a statement that one engine on one of its aircraft failed midflight on July 15 from Seattle, Washington to Austin, Texas, and the aircraft successfully made an emergency landing at the Austin airport. (84.5)
- **DPO(CP+TP):** This airline operator said in a statement that one of its aircraft was on its way from Seattle, Washington to Austin, Texas when one of its engines suddenly went out of order, and the aircraft successfully made an emergency landing at the airport in Austin. (81.4)
- **DPO-RK(CP+TP):** This airline operator said in a statement that one engine on one of its aircraft failed during the flight from Seattle, Washington to Austin, Texas on July 15, and the aircraft successfully made an emergency landing at the Austin airport. (85.4)
- **DPO-D(CP+TP):** This airline operator said in a statement that one engine on one of its aircraft failed during the flight from Seattle, Washington to Austin, Texas on July 15, and the aircraft successfully made an emergency landing at the Austin airport. (85.4)

The translation produced by the DPO(CP+TP) system fails to capture the date of the event ("July 15"). Translations from DPO-RK(CP+TP) and DPO-D(CP+TP) are identical and differ from that from DPO(CP) only in rephrasing "midflight" with "during the flight."

**Case 3: ID = 91**

**Source:** 运动少年热血竞技秀《运动吧少年》在7月11日正式登陆湖南卫视。

**Reference:** The inspiring competitive show for youth sports, We Are The One, was officially broadcast on Hunan TV on July 11.

- **DPO(CP):** The youth sports action show "Be Active, Be Sporty" will officially debut on Hunan Satellite TV on July 11. (76.8)
- **DPO(CP+TP):** The youth sports competition show "Go, Boy!" officially landed on the Changsha Satellite TV Station on July 11. (75.6)
- **DPO-RK(CP+TP):** The youth sports competition show "Go Play Boys" will be officially broadcast on the Hunan Satellite TV on July 11. (72.1)
- **DPO-D(CP+TP):** The youth sports competition show "Let's Go Boys" will officially debut on the Hunan Satellite TV on July 11. (74.2)

In this case, all systems produced a more literal translation of the name of the TV show compared to the reference. Although the KIWI-22 scores for DPO-RK(CP+TP) and DPO-D(CP+TP) are lower, the translation qualities are in fact similar.

## J Broader Impacts

This paper presents work whose goal is to advance the field of Machine Learning. There are many potential societal consequences of our work, none of which we feel must be specifically highlighted here.

