# OpenReview forum: "On Extending Direct Preference Optimization to Accommodate Ties"
_NeurIPS.cc/2025/Conference — NeurIPS 2025 poster_

### Official Review · Reviewer_87e1 · 2025-06-30

**Clarity:** 3
**Significance:** 3
**Originality:** 3
**Rating:** 5
**Confidence:** 3

**Summary:**

In this paper, the authors extend DPO with the ability to model ties (i.e., cases where the winner and loser are equally good or comparable). As the authors convincingly argue, their approach is a more principled alternative to simply removing such cases, which is a common pre-processing step in DPO due to known empirical disadvantages of having tied data instances. (though not mentioned, ties, or what are called "indifference relations" ~ are a well study theme in the broader literature on preference, see [Hansson, Sven Ove and Till Grüne-Yanoff, 2024]. The authors might consider citing this work as additional motivation).

To derive their variant of DPO, they rely on two established variants of Bradley-Terry (BT) that accomodate ties (named R(ao)K(upper) and D(avidson)). Both variants and very nicely summarized in Sec. 2.2 and (loosely speaking) define a distribution over w > l, l > w and a tie w ~ l via additional parameters (the difference in the two parameterizations of RK and D are discussed and explained from first principles in lines 74-87). They then use these two extensions to straightforwardly derive a general DPO variant (Eq. 14), one that factors into two loss terms that compete against one another, a term for clear winners with a binary weight and a term for ties with the same binary weight (both parameterized by either RK or D, giving rise to DPO-RK and DPO-D). Importantly, this general approach requires having an annotated set of tie and non-tie instances. (additional gradient analysis is provided in 2.3.1 that helps give intuition for how this differs from vanilla DPO).

Experiments are performed for machine translation, text summarization and math reasoning. Ties and non-ties are obtained heuristically using outside models to score and rank outputs (the details vary depending on each task, though are easy to understand). The issue of DPO with tied instances is shown clearly in Figures 1-2 and compared against a filtering-based approach (both serve as natural baselines for their new loss). Modest improvements are obtained for both MT (Table 1) and math reasoning over DPO trained without tied instances (my main concern relatses to the narrowness of these tasks and the modesty of the improvements). Additional analysis is done (all in the appendix) to show that their approaches yield higher reward accuracy, enforce improved regularization (a technical argument for this is provided 3.3) and has improved convergence behavior (given limited time, I only had a cursory view of these results. I would encourage the authors to move the most important parts of this into the main text).

====

Hansson, Sven Ove and Till Grüne-Yanoff, "Preferences", The Stanford Encyclopedia of Philosophy (Winter 2024 Edition), Edward N. Zalta & Uri Nodelman (eds.), URL = <https://plato.stanford.edu/archives/win2024/entries/preferences/>.

**Questions:**

-- Was significance testing done Tables 1-2?

-- Did you extend your approach to any additional tasks?

-- Have you considered making the binary `t` terms in Eq 14 real-valued? I ask because I'm left wondering how your approach relates to recent attempts at interpolating between multiple losses, e.g., in RainbowPO [Zhao et al. ICLR 2025]

-- I'm also left wondering whether your approach could be easily extended to handle noise. Conceptually, ties seem to express a symmetry between the scores of the winner and loser, either they are equally "bad" (i.e., their probability should be driven towards zero) or equally "good" (i.e., their probability should be driven toward 1). From what I gather, your objective only accounts for the latter both not the former. Does this intuiton make sense?

**Ethical Concerns:**

["NO or VERY MINOR ethics concerns only"]

**Final Justification:**

My biggest concern related to the limited empirical analysis and focus on MT, though these concerns were convincingly addressed by the authors. I also appreciated the additional feedback during the rebuttal.  I have therefore decided to keep my scor high.

**Limitations:**

yes

**Paper Formatting Concerns:**

none.

**Quality:**

3

**Strengths And Weaknesses:**

# strengths

-- A well motivated variant of DPO that directly addressed the important issue of ties in preference alignment. One that has clear and interesting connections with established variants of BT. I could imagine others in this area building on this novel direction and their approach.

-- The paper is very well written and easy to follow.

# weakensses

-- (minor) The experiments are narrow, focusing only on MT, text summarization and math reasoning. I feel that these particular tasks are not sufficiently motivated and would like to see the authors address this. For example, it is not clear to me why MT is an interesting tasks wrt ties versus other tasks.

-- The improvements reported in Tables 1-2 over standard DPO with filtering (as well as the baseline in row 1 for Table 1) are minimal and it is unclear whether they are significant.

-- (minor) Their particular formulation in some ways makes the task more complicated by needing annotate tied data. It is unclear, outside of the tasks they chose, how such data can be reliably collected. I would like to see the authors address this, and also explain if they think that their approach could be extended so that this portion is learned.

---

> ### Author Rebuttal · Authors · 2025-07-30
>
> Thank you for your careful and knowledgeable review, in response:
>
> > As the authors convincingly argue, their approach is a more principled alternative to simply removing such cases, which is a common pre-processing step in DPO due to known empirical disadvantages of having tied data instances. (though not mentioned, ties, or what are called "indifference relations" ~ are a well study theme in the broader literature on preference, see [Hansson, Sven Ove and Till Grüne-Yanoff, 2024]. The authors might consider citing this work as additional motivation).
>
> Thanks for pointing out this work. We will include it in our discussion.
>
> > It is not clear to me why MT is an interesting tasks wrt ties versus other tasks.
>
> Thanks for raising this. MT is actually a great task for studying ties, given how many ways there are to generate good translations.  The natural variability in translation means that ties are everywhere. Compared to summarization and general chat completion, Machine Translation also offers a range of well-established automatic metrics for assessing translation quality (e.g., BLEURT, XCOMET, KIWI-XXL), which we leverage to identify pairs of similar qualities as tied pairs reliably.    We will make this clear in our revision. We also note that applying DPO and its variants to MT is an established research topic with many published works. We follow them in designing our experiments.
>
> > Did you extend your approach to any additional tasks?
>
> We conduct experiments on three Machine Translation datasets (WMT, IWSLT, ALMA-R), one summarization dataset (TL;DR), and one mathematical reasoning dataset (GSM8K). Our findings are consistent among these tasks experimented. We chose these tasks and datasets to cover different setups where ties may arise: in MT multiple reliable automatic metrics are available; in TL;DR an external metric is not used for identifying TPs and we use the base DPO reward model; in GSM8K the response labels are binary (correct or false). We believe our experiments so far cover a good range of scenarios and applications.
>
> > The improvements reported in Tables 1-2 over standard DPO with filtering (as well as the baseline in row 1 for Table 1) are minimal and it is unclear whether they are significant. Was significance testing done Tables 1-2?
>
> Thanks for raising this.
>
> For Table 1, our baseline is directly comparable to Table 10 (zh sub-table) in the original CPO paper [Xu et. al, 2024]. They established that a +1 gain in KIWI-XXL over the baseline achieves 80% accuracy with human judgement of improvements in quality. Our DPO-RK(CP+TP) and DPO-D(CP+TP) models give >+1.4 gain in KIWI-XXL compared to the DPO(CP) model. We will run additional experiments to report on the stability of these gains with different random seeds, but note that in Machine Translation consistent gains across multiple recent metrics are generally taken as significant.
>
> The results from Table 2 are obtained by using the same random seed. We have run experiments for five additional random seeds for each variant at each beta value (75 training runs in total). In the table below, we report the median accuracy and standard deviation. We find that consistent with our previous finding, tie-compatible variants achieve higher median accuracy at all beta values. Gains are generally greater than 1x standard deviation.
>
>
>
> | $\beta$ | DPO(CP)         | DPO-RK(CP+TP)    | DPO-D(CP+TP)    |
> |--------:|------------------|-------------------|------------------|
> | 0.1     | 80.4% $\pm$ 1.34% | **82.9%** $\pm$ 0.90% | 82.2% $\pm$ 1.95% |
> | 0.3     | 83.7% $\pm$ 0.45% | **84.3%** $\pm$ 0.18% | 83.7% $\pm$ 0.68% |
> | 0.5     | 83.5% $\pm$ 0.14% | 84.1% $\pm$ 0.08% | **84.2%** $\pm$ 0.15% |
> | 0.7     | 83.6% $\pm$ 0.53% | 83.4% $\pm$ 0.19% | **84.5%** $\pm$ 0.58% |
> | 1.0     | 83.8% $\pm$ 0.29% | **84.0%** $\pm$ 0.08% | 83.7% $\pm$ 0.42% |
>
> We have also run experiments with IPO, CPO, and SimPO using the same CP set as used by DPO. We find that these variants generally yield inferior performance to DPO under the same random seed used for Table 2. Please kindly see details in our response to Reviewer Xv6R. This shows that it is non-trivial to obtain consistent gains over DPO(CP), which remains a hard-to-beat baseline.
>
> We also verify that the best-performing DPO-RK(CP+TP) and DPO-D(CP+TP) achieves a higher preservation rate compared to DPO(CP) (97.22%, 97.54% versus 96.04%). These numbers again agree with our results in Table 3.
>
> We emphasize that the main contribution of Sec.3.5.2 is to show that ties can be constructed and used in DPO-RK/D training to preserve good behaviour in the reference model through regularization. It is a novel approach and in our experiments on GSM8K, we find small yet consistent gains in overall performance and higher overall preservation rate from tie-regularized systems.
>
> > Their particular formulation in some ways makes the task more complicated by needing annotate tied data. It is unclear, outside of the tasks they chose, how such data can be reliably collected. I would like to see the authors address this, and also explain if they think that their approach could be extended so that this portion is learned.
>
> Thanks for your questions. We have demonstrated four ways to construct tied pairs: (1) via external automatic metrics (Sec.3.2, NMT); (2) via the trained DPO reward model (Sec.3.2; TL;DR); (3) via identifying pairs of conflicting preference annotations in existing pairwise datasets (Sec.3.5.1); and (4) via identifying pairs where both responses are correct when the label is binary (Sec.3.5.2). Though not exhaustive, we believe these offer a good range of approaches to tackle other downstream tasks. We will include the above discussion in our revision to improve clarity. We note Ameli et. al. [ICLR, 2024] has studied using ties in evaluating LLM chatbots.  They find that "the ability to handle ties—an integral aspect of human-judged comparisons—significantly improving the model’s fit to observed data". As we note in our introduction,  current development efforts collect and identify ties, but then discard them.  Our proposed approach simply requires keeping ties rather than throwing them out.
>
> > Have you considered making the binary t terms in Eq 14 real-valued? I ask because I'm left wondering how your approach relates to recent attempts at interpolating between multiple losses, e.g., in RainbowPO [Zhao et al. ICLR 2025]
>
> Thanks for raising this insightful point. Our focus in this paper is accommodating ties in DPO when tied pairs are explicitly declared and so we keep the indicator $t$ binary-valued. Although we have not explored this, we note that the effect of a continuous $t$ on the gradients is relatively straight-forward: it would encourage a small margin between the winning and losing elements of tied pairs.
>
> Interpolating between multiple losses is an interesting topic and RainbowPO does offer a unified framework for many DPO variants.  However their formulation (their eq 18) assumes that expectation is carried out over clear preferences (i.e. no ties).    Conceivably, it would be possible to work from Eq.18 in the RainbowPO paper to extend the expectation to include ties, analogous to what we have done in Section 2.3 for plain DPO.
>
> > I'm also left wondering whether your approach could be easily extended to handle noise. Conceptually, ties seem to express a symmetry between the scores of the winner and loser, either they are equally "bad" (i.e., their probability should be driven towards zero) or equally "good" (i.e., their probability should be driven toward 1). From what I gather, your objective only accounts for the latter both not the former. Does this intuiton make sense?
>
> Thanks for your question. Our objective accounts for both cases as it aims to assign a high probability of declaring a tie on labeled Tied Pairs (TP). That is, the objective does not distinguish between whether the two responses are good or bad in terms of absolute quality.
>
> Indeed, the approach could be extended to handle noisy comparisons. One such scenario is when multiple annotators give conflicting judgements, which we experiment in Sec 3.5.1 and  show including these data as tied give performance gain compared to discarding them. More generally, developers could choose to model pairs that are not certain in preference direction, owing to annotation noise, metric precision or otherwise, as ties. We believe our proposed variants and findings can inform future research in this direction.
>
> **References**
>
> Siavash Ameli, Siyuan Zhuang, Ion Stoica, and Michael W. Mahoney. A statistical framework for ranking LLM-based chatbots. In The Thirteenth International Conference on Learning Representations, 2025.

---

> > ### Comment · Reviewer_87e1 · 2025-08-04
> >
> > Thank you for the detailed feedback. In particular, the points you make about my wuestion "why MT?" is compelling, as are the additional experiments you reported above.
> >
> > I will adjust my score, since my main concerns were (I feel) appropriately addressed.
> >
> > > Thanks for your question. Our objective accounts for both cases as it aims to assign a high probability of declaring a tie on labeled Tied Pairs (TP). That is, the objective does not distinguish between whether the two responses are good or bad in terms of absolute quality.
> >
> > Here's a more concrete question. Suppose that both losses are tied and they are both good generations (i.e., it make sense to drive their probability up towards probability 1.0). A natural loss for this might just be $-log( p(w) * p(l) )$ (this ends up being effectively a CE loss). In contrast, both could be bad, in which case you might want to (intuitively) optimize for unlikelihood $-\log (1 - p(w) * 1 - p(l))$`. Do these two cases follow as special cases under your parameterization?

---

> > > ### Author Response · Authors · 2025-08-05
> > >
> > > Thank you for your recognition. We are glad that you find the response informative, and will make sure that these experiments and discussion you inspired are incorporated in our revision.
> > >
> > > > Do these two cases follow as special cases under your parameterization?
> > >
> > > Thanks for raising this. The (un)likelihood losses you specified do not include terms from the reference models, and so are not captured by DPO and our DPO-based variants. This points to a somewhat counter-intuitive property that DPO only works to increase reward margins (i.e., difference of policy-reference likelihood ratio), but not necessarily the individual likelihoods.
> > >
> > > However, the intuition is valid, in that it may be desirable to increase/decrease likelihood of tied pairs based on their absolute quality. Adding an additional likelihood-based criterion as you proposed to DPO-RK and DPO-D could accommodate this. In contrast, modification over the original DPO would struggle to accommodate this as there is no concept of ties.

---

> ### Comment · Reviewer_87e1 · 2025-08-05
>
> > The (un)likelihood losses you specified do not include terms from the reference models, and so are not captured by DPO and our DPO-based variants.
>
> Right, this is a good point, but nonetheless, we can consider the special *(and, in practice, unnatural)* case when the reference win/lose probabilities ($W_{ref}, L_{ref}$) are **equal**  to cancel out the reference terms (conceptually, this is the case that is relevant when considering ties since the probabilities should be nearly equal).
>
> I.e., through algebra we can write DPO (without $\beta$) as $$-\log \sigma \bigg( \log \frac{W}{L} - \log \frac{W_{ref}}{L_{ref}} \bigg)$$ to see that the last *reference policy* term goes away under such a case (this form of DPO is often used to compare standard DPO with reference-free approaches like SimPO, which replaces the second term with a parameter $\gamma$).
>
> Using this idea, I can therefore write a DPO-style likelihood loss as $$-\log \sigma \bigg(\log \frac{WL}{(1 - (WL))} - \log \frac{W_{ref}}{L_{ref}}  \bigg)$$which (as above) equals $-\log ( WL)$ when $L_{ref} = W_{ref}$. (*through a similar process an analogous unlikelihood loss could be defined.*)
>
> The point is that the loss above seems reasonable when dealing with ties where the winner and loser are both **good** (i.e., it make sense to maximize the probability of both winner and loser). Given this, I would hope that this kind of loss follows as a special of your loss, but it's a little hard to see given the derivation from these complex forms of BT.
>
> (**note**: I am just asking out of curiousity)

---

> > ### Author Response · Authors · 2025-08-06
> >
> > > I would hope that this kind of loss follows as a special of your loss, but it's a little hard to see given the derivation from these complex forms of BT.
> >
> > Thanks for sketching the derivation and your continued engagement. We will study this idea as extensions of our work. If we see a good way to implement your idea via simple modifications over our losses, we will include a discussion in the appendix.
> >
> > We note that another way to maximize probabilities of both responses in TPs (in case they are good) is via special CPs/TPs construction. Consider the case of three responses: $y_1 \sim y_2 \succ y_3$. If we set $(y_1, y_3)$ as CP and $(y_1, y_2)$ as TP, then we can expect the rewards corresponding to $y_1$ and $y_2$ ($r_1$ and $r_2$) to increase. This is because through the $(y_1, y_3)$ CP, $r_1$ increases and through the $(y_1, y_2)$ TP, $r_2$ should approach $r_1$. This should apply to both DPO-style or SimPO-style (reference-less) approaches, and so your special case based on log-likelihood can be incorporated. The unlikelihood case can be accommodated similarly by reversing the preference direction in CP.
> >
> > Thanks again for raising this. We believe this is a case where the introduction of TPs provides a convenient tool to model edge cases that developers may care about. While the focus of this work is to establish the theoretical framework that admits TPs and show some ways to construct TPs, we look forward to exploring more modeling techniques in future work.

---

> ### Comment · Reviewer_87e1 · 2025-08-06
>
> > We note that another way to maximize probabilities of both responses in TPs (in case they are good) is via special CPs/TPs construction....
>
>
> This is very interesting, thank you for these additional details.
>
>
> > We will study this idea as extensions of our work.
>
> **As a final thought**: I recognize that the losses I sketch above are quite strange from the perspective of DPO and BT. It does feel like deriving these variations via BT and classical RLHF, though reasonable and consistent with the original DPO derivation, does sometimes get in the way of understanding. You might consider (if you haven't already) the kind of approach of [Tang et al. ICML 2023] that avoids an explicit BT parameterization; [Richardson et al. ICML 2025] generalizes this approach in a way that allows one to derive the kinds of losses mentioned above.
>
> *[Tang et al. ICML 2024]  Generalized Preference Optimization: A Unified Approach to Offline Alignment*
>
> *[Richardson et al. ICML 2025] Understanding the logic of direct preference alignment through logic.*

---

> > ### Author Response · Authors · 2025-08-09
> >
> > Thank you for your knowledgable review and providing us with these references. We discuss Tang et al. in Sec.5 Related Work and will examine Richardson et al. closely to understand if there is an opportunity to build on their approach as extensions of this work. Thanks again for your suggestions.

---

### Official Review · Reviewer_p1pE · 2025-07-01

**Clarity:** 3
**Significance:** 3
**Originality:** 2
**Rating:** 4
**Confidence:** 4

**Summary:**

While Bradley-Terry model assumes preferences have clear winners and losers, this is not the case in practice. In fact, previous work ignored data points that represent ties. This paper extends DPO to accommodate ties using two existing approaches: Rao&Kupper and Davidson. These models differ in how they assign probabilities to win and tie cases. They also introduce additional parameters that control the mass assigned to ties. Analyzing the gradients, these two variants (DPO-RK and DPO-D) exhibit different scaling factors in gradient estimation. Experimental results on NMT and summarization suggest that (i) adding ties have a regularization effect, (ii) DPO-RK and DPO-D achieves better KL vs performance curves, (iii) while DPO reward estimation is better for win/loss, other variants offer better reward estimation for ties.

**Questions:**

1. Please add more preference learning baselines.

2. Please develop simple DPO baselines to incorporate ties.

3. Please clarify how TP dataset is created.

4. Please run more experiments with DPO-RK (CP) and DPO-D (CP).

5. Would DPO serve as a better reward modeling for near-ties based on Figure-3?

**Ethical Concerns:**

["NO or VERY MINOR ethics concerns only"]

**Final Justification:**

Clarifications and additional results.

**Limitations:**

Yes

**Quality:**

2

**Strengths And Weaknesses:**

# Strengths
1. The paper is well-written and easy-to-follow. It provides a clear motivation on why ties could help and successfully points to existing work in utility theory for the proposed problem.

2. The authors introduce two extensions: DPO-RK and DPO-D. Both extensions are clearly explained and connected to existing baseline DPO model.

3. Experimental results are conducted on real-world benchmarks (NMT and summarization) with KL-performance curves and ablations on $\beta$ parameter.

# Weaknesses
1. **Lack of baselines.** There exists a large body of preference optimization objectives that add margins, different losses, etc. that are not studied in the paper. While the introduced variants show some promise, it is not clear if they are already captured by the existing DPO variants in the literature.

2. **Simple DPO baselines for ties.** Can you run experiments with baseline DPO where different weights are assigned to win/loss or ties? Is it that introduced variants capture ties better or can a simple extension of DPO also work well in practice?

3. **Clarification on TP dataset.** When you are creating the TP dataset, are you using only one direction for ties? I.e., given $y_1$ and $y_2$ is a tie, is only one of $y_1 > y_2$ or $y_1 < y_2$ present in the TP dataset for DPO? If it is the case, how would adding both directions impact DPO performance?

4. **DPO-x ablations.** Can you train DPO-RK and DPO-D on CP-only data? It is not clear if extending the dataset with ties or the slightly different preference modeling is why these variants work better.

5. **Clarification on Figure-3.** In Figure-3 (left), DPO achieves close to zero preference probability when $d_\theta$ is also close to zero. Does this mean that, when used as a reward model to rank candidates, DPO performs better for "almost ties"?

6. **Minor typos.** In Eq (23), $\beta$ should be removed from the RHS.

---

> ### Author Rebuttal · Authors · 2025-07-30
>
> Thanks for your careful and knowledgable review. In response:
>
> > it is not clear if they are already captured by the existing DPO variants in the literature.
>
> Thanks for raising this. We discuss related variants (e.g., ODPO) and our theoretical contribution compared to previous work in Related Work (Sec.5). To our knowledge, there are no other DPO variants that explicitly model ties except for contemporaneous work by Guo et. Al. [2024], which we compare in detail in Sec.5.
>
> > Can you run experiments with baseline DPO where different weights are assigned to win/loss or ties? Is it that introduced variants capture ties better or can a simple extension of DPO also work well in practice?
>
> Thanks for raising this.
>
> In fact, the introduced variants do capture ties better from a modeling perspective. Considering the alternative which would be to assign different weights to win-loss and ties, the DPO objective would still try to increase the margins on ties, which is suboptimal from a modeling perspective because ties are taken to be equivalent. In Sec.2.3.1., we show that the DPO-RK and DPO-D gradients can work to decrease margins on ties, which is a mechanism not present in DPO, even with weights. A simple extension of DPO may work well in practice, but would struggle to accommodate ties.
>
> > Lack of baselines. There exists a large body of preference optimization objectives that add margins, different losses, etc. that are not studied in the paper.
>
> Thanks for raising this. We choose to focus on a direct comparison with DPO in the main experiments. This is to allow us to directly study the effect of replacing Bradley-Terry with tie-compatible comparison models without interference from other modifications (e.g., length normalization, removing reference model, etc). However, we agree that results from other PO methods could provide useful points of comparison for interested researchers. We have run three established PO methods (IPO, SimPO, and CPO) on the GSM8K dataset using the same CP set as used by DPO. The results are tabulated below:
>
>  | β   | SimPO (CP) | CPO (CP) | IPO (CP) | DPO (CP) | DPO-RK (CP+TP) | DPO-D (CP+TP) |
> |-----|------------|----------|----------|----------|----------------|----------------|
> | 0.1 | 82.5%      | 82.6%    | 20.2%    | 76.4%    | **83.5%**          | 81.7%          |
> | 0.3 | 81.8%      | 83.1%    | 57.1%    | 83.7%    | **84.4%**          | 83.2%          |
> | 0.5 | 81.8%      | 83.1%    | 69.1%    | 83.6%    | 83.8%          | **84.5%**          |
> | 0.7 | 81.6%      | 82.6%    | 78.3%    | 83.3%    | 83.7%          | **84.5%**          |
> | 1.0 | 82.2%      | 83.6%    | 76.1%    | 83.5%    | **84.1%**          | 83.7%          |
>
> We find that in all but one case (CPO(CP), $\beta=1.0$) these variants give lower accuracy compared to DPO. DPO-RK(CP+TP) and DPO-D(CP+TP) still yield the best performing system at every beta value with the added baselines. We believe these add to the strength of our empirical evidence.
>
> > When you are creating the TP dataset, are you using only one direction for ties? I.e., given $y_1$ and $y_2$ is a tie, is only one of  $y_1 > y_2$ or $y_2 > y_1$ present in the TP dataset for DPO? If it is the case, how would adding both directions impact DPO performance?
>
> In our experiments on WMT, IWSLT and TL;DR, we always provide the higher scoring response as the chosen response when training with DPO (Line 786-787, Appendix E.1).  We did not add two training instances for one pair (i.e., provide both directions) because the gradients would cancel as we provide pairs for the same prompt in the same batch following the original implementation of DPO by Rafailov et al. [2023].
>
> We will include experiments where both directions are provided to DPO and not in the same batch. We have some intuitions from our findings so far: in Appendix F.6., we reverse the preference direction of Tied Pairs (rTP) and find that DPO(CP+TP) and DPO(CP+rTP) give similar performance (58.6% versus 60.9%). Both are substantially lower than DPO-RK(CP+TP) and DPO-D(CP+TP) (68.0% and 68.8%); In Appendix F.3., we find a higher proportion of TPs in training data leads to enhanced regularization effect. From these experiments, it is likely that providing both directions would not result in substantial performance gain but stronger regularization.
>
> > Can you train DPO-RK and DPO-D on CP-only data? It is not clear if extending the dataset with ties or the slightly different preference modeling is why these variants work better.
>
> Thanks for raising this. We will include DPO-RK(CP) and DPO-D(CP) results in our final version for completeness. We emphasize that our variants are specifically designed to handle cases where Tied Pairs are present. As we have shown with the additional baselines, DPO remains a hard-to-beat method when only CPs are available and it is unlikely that replacing the Bradley-Terry model with Rao-Kupper or Davidson without providing Tied Pairs would enhance performance.
>
> > In Figure-3 (left), DPO achieves close to zero preference probability when $d_\theta$ is also close to zero. Does this mean that, when used as a reward model to rank candidates, DPO performs better for "almost ties"? ... Would DPO serve as a better reward modeling for near-ties based on Figure-3?
>
> Thanks for raising this. Figure 3 (left) shows the clear preference probabilities $P(y_w ≻ y_l |x)$ (left) as a function of reward margins $d_θ(x, y_w, y_l)$ for Bradley-Terry (See caption). When the reward margin is close to zero, the preference probability in either direction is close to 0.5. Note that different from the Rao-Kupper and the Davidson model, the Bradley-Terry model never explicitly assigns probability to the event of ties. In other words, a decision-rule based on the predicted margin is required for the DPO model to classify pairs as ties.
>
> In Appendix C.2, we find that the DPO(CP) model, when used as a reward model, performs especially poorly at classifying tied pairs on held-out datasets compared to DPO-RK and DPO-D models (>20% difference in accuracy). This is true under two decision rules detailed in Appendix C.2. The DPO(CP+TP) model achieves higher accuracy at classifying TPs but at the cost of substantial degradation in CPs classification. We find that DPO-RK(CP+TP) and DPO-D(CP+TP) models achieve higher overall classification accuracies under both decision rules compared to DPO models.
>
> >  In Eq (23), should be removed from the RHS.
>
> Thanks for catching this! We will fix this in our revision. Many thanks.

---

> > ### Comment · Reviewer_p1pE · 2025-08-08
> >
> > Thank you for the clarifications and additional results. I have updated my score accordingly.

---

> > > ### Author Response · Authors · 2025-08-09
> > >
> > > Thanks again for your careful review and recognition. We will make sure to incoporate these into our revision.

---

### Official Review · Reviewer_eagA · 2025-07-01

**Clarity:** 2
**Significance:** 3
**Originality:** 3
**Rating:** 4
**Confidence:** 4

**Summary:**

Direct Preference Optimization (DPO) effectively aligns large language models (LLMs) by maximizing the log-likelihood of preference samples, offering a simpler alternative to traditional reinforcement learning (RL) pipelines. However, the original DPO, which relies on the Bradley-Terry model, cannot handle "tie" outcomes common in real-world preference data, which are typically discarded, leading to information loss and performance degradation. To address this, this paper proposes two DPO extensions that support tie modeling: DPO-RK (based on the Rao-Kupper model) and DPO-D (based on the Davidson model), which explicitly model tie outcomes in pairwise comparisons. Experiments demonstrate that both methods significantly outperform the original DPO in translation, summarization, and mathematical reasoning tasks. Furthermore, the inclusion of tie data enhances model stability and generalization capability by providing stronger regularization.

**Questions:**

**Is Discarded "Tie" Data Necessarily High-Quality and Valuable Information?** The paper posits "tie data is typically discarded, leading to information waste" (e.g., Introduction, Lines 30-32) as one of its core motivations. However, the robustness of this premise is debatable: In real-world preference data, instances labeled as "ties," beyond genuine preference ambiguity, might also include data that is inherently low-quality, difficult to distinguish in terms of preference, or stems from annotation noise. Could the authors provide stronger evidence or argumentation to support the claim that the "tie" data repurposed by their methods generally possesses high informational value? This would be necessary to substantiate the universality and urgency of the "avoiding waste" motivation, rather than potentially introducing unnecessary complexity or interference in certain scenarios.

**Ethical Concerns:**

["NO or VERY MINOR ethics concerns only"]

**Final Justification:**

After the rebuttal, most of my concerns are well addressed. I think this paper can be accepted

**Limitations:**

yes

**Quality:**

3

**Strengths And Weaknesses:**

## Strengths
**Innovative and Effective DPO Extensions to Address Wasted Tie Preference Data**: Addressing the practical limitation of DPO's inability to directly handle "tie" preference data, the paper innovatively proposes two variants, DPO-RK and DPO-D. These variants explicitly model tie probabilities by integrating the Rao-Kupper and Davidson probabilistic models, respectively. Experiments demonstrate that these methods, when incorporating tie data, can prevent performance degradation and enhance model performance.
Thorough Validation of the Proposed Methods' Practical Value in Enhancing Model Performance and Utilizing Discarded Data: Through extensive experiments across multiple tasks such as machine translation, summarization, and mathematical reasoning, the paper demonstrates that DPO-RK and DPO-D can effectively utilize tie data previously discarded due to ambiguity or conflict. Consequently, they achieve superior performance compared to strong DPO baselines on these tasks and provide valuable strategies for handling and selecting tie data in practical applications.

## Weaknesses
**Neglect of β's Influence in α Parameter Sensitivity Analysis**: Although Appendix F.1 explores the sensitivity of α_RK and ν_D parameters, the experiments were conducted using a single β value (e.g., β=0.4 for DPO-RK and β=0.5 for DPO-D). The analysis does not investigate whether variations in β would affect the sensitivity results. Given that β is a core regularization strength parameter in the DPO framework, its value directly influences the model's learning from positive/negative preference samples. Therefore, when investigating the performance impact of α parameters (which control tie-handling strategies), neglecting β might obscure potential interaction effects between α and β. It could also lead to an incomplete understanding of the model's hyperparameter tuning mechanism.

**Inconsistent β Values Between Sensitivity and Main Experiments without Justification**: The β values used in the α parameter sensitivity experiments in Appendix F.1 are inconsistent with those used in the main experiments (e.g., Figures 1 and 2) where the model achieved optimal performance. No reasonable explanation for this choice is provided. This inconsistency raises questions: Why wasn't the sensitivity analysis of α parameters conducted at the β values that yielded optimal model performance? Could different choices of β alter the optimal values of α? The paper does not discuss this, which reduces confidence in the conclusions drawn from the sensitivity analysis.

**Ambiguous Definition of Hyperparameter α in Sensitivity Analysis**: In the main text, the paper explicitly substitutes the tie-control parameter ν_RK in DPO-RK with α_RK. However, it does not provide a similar substitution or equivalent expression for ν_D in DPO-D. Yet, in the sensitivity analysis in Appendix F.1, α is uniformly used to represent the tie-control parameter for both methods without necessary clarification or consistent symbolic linkage. This ambiguous definition and opaque symbol substitution can easily cause confusion.

---

> ### Author Rebuttal · Authors · 2025-07-30
>
> Thank you for your positive and insightful review. In response:
>
> > Appendix F.1 explores the sensitivity of α_RK and ν_D parameters, the experiments were conducted using a single β value... Could different choices of β alter the optimal values of α?
>
> Thanks for raising this and your careful review. We find $\alpha_{RK}=ln3$ and $\nu_D=1$ work well for the range of beta values (0.1-0.9) tested, but agree that there is an opportunity to finetune the hyper-parameter for optimal performance as we discuss in the Limitations section (Line 658-662). We have conducted a grid search of beta and alpha values on WMT to investigate this. Please see the table below for results:
>
> | Variant          | $\beta$ | $\alpha_{RK}$ | KL    | COMET |
> |------------------|---------|---------------|-------|--------|
> | DPO-RK(CP+TP)     | 0.2     | 0.1           | 19.73 | 80.9   |
> |                  |         | 0.5           | 22.31 | 81.0   |
> |                  |         | ln3           | 28.61 | 80.9   |
> |                  |         | 2.0           | 37.05 | **81.1**   |
> |                  |         | 3.0           | 49.00 | 81.0   |
> | DPO-RK(CP+TP)     | 0.4     | 0.1           | 9.54  | 80.9   |
> |                  |         | 0.5           | 14.80 | 81.0   |
> |                  |         | ln3           | 21.84 | **81.1**   |
> |                  |         | 2.0           | 30.41 | **81.1**   |
> |                  |         | 3.0           | 76.66 | 74.6   |
> | DPO-RK(CP+TP)     | 0.8     | 0.1           | 6.32  | 80.7   |
> |                  |         | 0.5           | 5.91  | 80.6   |
> |                  |         | ln3           | 8.10  | 80.8   |
> |                  |         | 2.0           | 10.37 | 80.9   |
> |                  |         | 3.0           | 14.53 | 80.9   |
>
>
> | Variant          | $\beta$ | $\alpha_{D}$ | KL    | COMET |
> |------------------|---------|--------------|-------|--------|
> | DPO-D(CP+TP)     | 0.2     | -2.5         | 27.46 | 80.9   |
> |                  |         | -1.0         | 34.80 | 80.6   |
> |                  |         | 0            | 42.74 | 80.8   |
> |                  |         | 1.0          | 56.83 | 80.6   |
> | DPO-D(CP+TP)     | 0.4     | -2.5         | 14.56 | 81.0   |
> |                  |         | -1.0         | 18.71 | 81.0   |
> |                  |         | 0            | 17.01 | 81.0   |
> |                  |         | 1.0          | 33.15 | 81.0   |
> |                  |         | 2.5          | 56.18 | 80.7   |
> | DPO-D(CP+TP)     | 0.5     | -2.5         | 11.56 | 81.0   |
> |                  |         | -1.0         | 15.25 | **81.1**  |
> |                  |         | 0            | 11.66 | 81.0   |
> |                  |         | 1.0          | 29.72 | 81.0   |
> |                  |         | 2.5          | 47.20 | **81.1**   |
> | DPO-D(CP+TP)     | 0.8     | -2.5         | 8.16  | 80.7   |
> |                  |         | -1.0         | 9.10  | 80.7   |
> |                  |         | 0            | 13.33 | **81.1**   |
> |                  |         | 1.0          | 19.54 | 81.0   |
> |                  |         | 2.5          | 30.48 | 80.8   |
>
> Consistent with our findings in Appendix F.1., we find that the task performance is stable over a range of $\alpha_{RK}$ and $\alpha_{D}$. These results add more empirical evidence to the role of $\alpha_{RK}$ and $\alpha_{D}$ as a knob for regularization strength. We see that, controlled for a fixed value of $\beta$, increasing $\alpha_{RK}$ and $\alpha_{D}$ generally leads to higher KL with respect to the reference model at the end of training. The gradient-based analysis in Appendix F.1. could explain this result (Line 885-887): "Given large α, the gradient on TPs approach zero regardless of the reward margin of the pair under the current model. As we show in the paper, tied pairs contribute to a regularization effect".
>
> Given these experiments, we suggest the following scheme in hyper-parameter tuning in DPO-RK and DPO-D: (1) run training with a range of beta values with the default $\alpha_{RK}=\ln 3$ and $\alpha_{D}=0$ to identify the working range of beta; (2) increase or decrease $\alpha_{RK}=\ln 3$ and $\alpha_{D}=0$ at those beta values, using the final KL as a guide. Note that as we show in experiments there is usually a "sweet range" of KL where performance is optimal. This procedure would allow developers to find the best performance-KL trade-off on a particular setup.
>
> Thanks again for suggesting this. We believe this adds to the practicality of our methods. We will include these in Appendix F.1 and mention our finding in the main text in our revision.
>
> > in the sensitivity analysis in Appendix F.1, α is uniformly used to represent the tie-control parameter for both methods without necessary clarification or consistent symbolic linkage.
>
> Thanks for raising this. We will make clear that $\alpha_{RK}=\log\nu_{RK}$ and $\alpha_D = \log \nu_D$ are defined for DPO-RK and DPO-D in Appendix F.1, respectively. Thanks for pointing this out.
>
> > The paper posits "tie data is typically discarded, leading to information waste" (e.g., Introduction, Lines 30-32) as one of its core motivations...... Could the authors provide stronger evidence or argumentation to support the claim that the "tie" data repurposed by their methods generally possesses high informational value?
>
> Thanks for raising this. We take the view that the informational value of tied pairs depends heavily on how pairs are collected and constructed. In this paper, we experiment on three common scenarios where ties naturally occur: (1) when pairs are similarly ranked/scored; (2) when pairs show conflicting preference from different annotators; and (3) when pairs contain responses that are both correct. We show that tied pairs contribute to strong regularization effect and could lead to improved performance over system trained with clear preferences only. This supports that these otherwise discarded data contain useful information for the learning procedure.
>
> While it is difficult to determine whether ties generally possess high informational value, we believe our work offers useful methodology and empirical evidence to investigate this question on a case-by-case basis.
>
> Outside DPO, we know that modeling ties is becoming increasingly important as overall performance improves. In evaluating recent LLMs,  Ameli. et. al [ICLR, 2025] finds that "the ability to handle ties—an integral aspect of human-judged comparisons—significantly improving the model’s fit to observed data". We will include this discussion in the revised version.
>
> **References**
>
> Siavash Ameli, Siyuan Zhuang, Ion Stoica, and Michael W. Mahoney. A statistical framework for ranking LLM-based chatbots. In The Thirteenth International Conference on Learning Representations, 2025.

---

> > ### Comment · Reviewer_eagA · 2025-08-05
> > **Response to Authors**
> >
> > Thanks for your responses, which help a lot. Please incorporate these points into your revisions.

---

> > > ### Author Response · Authors · 2025-08-05
> > >
> > > Thank you for your acknowledgement. We will make sure these points are incorporated in our revisions. We are grateful for your careful review which helps strengthen the paper.

---

### Official Review · Reviewer_Xv6R · 2025-07-06

**Clarity:** 3
**Significance:** 2
**Originality:** 2
**Rating:** 4
**Confidence:** 4

**Summary:**

The author observed that the original DPO does not allow for ties for LLMs’ alignment, i.e., does not allow for ambiguity in the comparison of the paired examples in the training data. The author thereby investigate DPO variants that can incorporate ties during training. The authors proposed two tie-compatible DPO variants, DPO-RK and DPO-D, by replacing the BT preference model with the Rao-Kupper and the Davidson models.

**Questions:**

1. Are there any other ways to accommodate ties in DPO?
2. Besides TPs, are there any other factors that have impact on the performance of DPO for LLMs’ alignment? Any discussions?

**Ethical Concerns:**

["NO or VERY MINOR ethics concerns only"]

**Limitations:**

The main limitation of the paper is that: this is an extension of the standard DPO model, which makes the contribution of the paper sounds marginal.

**Paper Formatting Concerns:**

NA.

**Quality:**

3

**Strengths And Weaknesses:**

Strengths and weaknesses:
1. The motivations of the proposed method is clear: to allow ties for LLMs’ preference alignment during training.
2. The writing of the paper is easy to follow.
3. Experiments were conducted to evaluate the performance of the proposed model.

Weaknesses of the paper:
1. The authors pair the top- and bottom-ranked responses as CPs and TPs as pairs that are close in the ranking. Not sure are there other way to construct CPs and TPs for training? Although the authors detailed the way they constructed the dataset for experiments, the constructed dataset was not publicly available, which making the reimplementation of the experiments impossible?
2. The paper suffers from the lack of novelty. The paper sounds like an incremental work the extension of existing Direct Preference Optimization, DPO. The main difference lies in the fact that the authors considered TPs in the objective.
3. Are there any other ways to accommodate ties in DPO?
4. There are some DPO variants being proposed in the literature. Other DPO variants should be taken into account as baselines in the experiments.

---

> ### Author Rebuttal · Authors · 2025-07-30
>
> Thanks for your positive and careful review. In response:
>
> > The authors pair the top- and bottom-ranked responses as CPs and TPs as pairs that are close in the ranking. Not sure are there other way to construct CPs and TPs for training?
>
> Thanks for your questions. We have demonstrated four ways to construct tied pairs: (1) via external automatic metrics (Sec.3.2, NMT); (2) via the trained DPO reward model (Sec.3.2; TL;DR); (3) via identifying pairs of conflicting preference annotations in existing pairwise datasets (Sec.3.5.1); and (4) via identifying pairs where both responses are correct when the label is binary (Sec.3.5.2). Though not exhaustive, we believe these offer a good range of approaches to tackle other downstream tasks.
>
> Expanding on (3) and (4) above, we show how to construct TPs other than based on ranking: by modeling conflicting preferences under multiple annotators as ties (ALMA-R-Preference, Machine Translation, Sec.3.5.1) and by modeling pairs that are both correct as ties (GSM8K, Mathematical Reasoning, Sec.3.5.2). We show that these approaches lead to improved performance over the DPO baseline without TPs. We will highlight these results in the introduction in revision to make clear we studied alternative ways to construct TPs.
>
> > Although the authors detailed the way they constructed the dataset for experiments, the constructed dataset was not publicly available
>
> Thanks for raising this point. We will publish the constructed CP+TP datasets upon publication. All datasets used have been statically saved to disk to ensure reproducibility.
>
> > The paper suffers from the lack of novelty..... The main difference lies in the fact that the authors considered TPs in the objective.
>
> Thanks for raising this. The DPO-RK/D formulation is one of our contributions. Other contributions include (1) To our knowledge, we are the first to show the regularization effect of including ties in DPO and its variants. We provide extensive empirical results and theoretical analysis to demonstrate this effect (Sec 3.1-3.4); (2) We show two novel ways that model previously discarded data as ties to improve model performance (Sec.3.5).  To our knowledge, this has not been investigated in the DPO literature.
>
> > Are there any other ways to accommodate ties in DPO?
>
> Thanks for raising this. We think this is an open research question, for which this paper derives and investigates two tie-compatible variants based on the Rao-Kupper and the Davidson models. To our knowledge, there are no other established methods to accommodate ties in DPO.
>
> > Other DPO variants should be taken into account as baselines in the experiments.
>
> Thanks for raising this point. We are specifically interested in the effect of replacing the Bradley-Terry model with tie-compatible comparison models in DPO to accommodate ties. Comparing directly to DPO in the main experiments allows us to study this replacement without interference from the modifications from other variants (e.g., length modifications and leaving out the reference model, etc.).
>
> We do provide IPO results in Appendix F.4., where we find including ties also lead to a slight degradation in performance and overall inferior performance to DPO. Related variants are discussed in Related Work (Sec.5).
>
> While not directly relevant to our investigation on ties, we agree that results from other DPO variants could serve as point of comparison for interested researchers. We have run three established PO methods (IPO, SimPO, and CPO) on the GSM8K dataset using the same CP set as used by DPO. The results are tabulated below:
>
>  | β   | SimPO (CP) | CPO (CP) | IPO (CP) | DPO (CP) | DPO-RK (CP+TP) | DPO-D (CP+TP) |
> |-----|------------|----------|----------|----------|----------------|----------------|
> | 0.1 | 82.5%      | 82.6%    | 20.2%    | 76.4%    | **83.5%**          | 81.7%          |
> | 0.3 | 81.8%      | 83.1%    | 57.1%    | 83.7%    | **84.4%**          | 83.2%          |
> | 0.5 | 81.8%      | 83.1%    | 69.1%    | 83.6%    | 83.8%          | **84.5%**          |
> | 0.7 | 81.6%      | 82.6%    | 78.3%    | 83.3%    | 83.7%          | **84.5%**          |
> | 1.0 | 82.2%      | 83.6%    | 76.1%    | 83.5%    | **84.1%**          | 83.7%          |
>
> We find that in all but one case (CPO(CP), $\beta=1.0$) these variants give lower accuracy compared to DPO. DPO-RK(CP+TP) and DPO-D(CP+TP) still yield the best performing system at every beta value with the added baselines. We believe these add to the strength of our empirical evidence.
>
> > Besides TPs, are there any other factors that have impact on the performance of DPO for LLMs’ alignment? Any discussions?
>
> Recent literature has studied many factors that could impact DPO performance, including length normalization, doing away with the reference model, token-wise formulation, etc. This is an on-going research topic and is more carefully discussed in recent survey papers (e.g., [Liu et al, 2025]). We note that the problem of accommodating ties remain under-investigated despite the large amount of literature. We discuss relevant DPO variants and frameworks in our Related Work section  (Sec. 5) to the best of our knowledge.
>
> **References**
>
> Shunyu Liu, Wenkai Fang, Zetian Hu, Junjie Zhang, Yang Zhou, Kongcheng Zhang, Rongcheng Tu, Ting-En Lin, Fei Huang, Mingli Song, Yongbin Li, Dacheng Tao: A Survey of Direct Preference Optimization.

---

### Comment · Area_Chair_Uxoz · 2025-08-06

Reviewers, please engage in the discussion. According to the policy this year, you must engage in the discussion before submitting the Mandatory Acknowledgement.. Otherwise, your review will be flagged as insufficient review.

---

### Decision · Program_Chairs · 2025-09-17

**Decision:**

Accept (poster)

**Comment:**

(a) Summary: The paper addresses a significant limitation of the original Direct Preference Optimization (DPO) for aligning Large Language Models (LLMs) -- its inability to accommodate "tie" outcomes in training data, which leads to information loss because such data is typically discarded. The traditional DPO, relying on the Bradley-Terry model, assumes clear winners and losers, a premise often not met in real-world preferences. To overcome this, the authors propose two novel DPO extensions that explicitly model tie outcomes: DPO-RK (based on the Rao-Kupper model) and DPO-D (based on the Davidson model). These variants introduce additional parameters to control the probability assigned to ties.

(b) Strengths:
- The paper effectively highlights the practical problem of DPO's inability to handle ties, providing a strong and clear motivation for the proposed method.
- The paper introduces innovative DPO extensions (DPO-RK and DPO-D) by integrating established probabilistic models (Rao-Kupper and Davidson) to explicitly model tie probabilities. This is considered a more principled alternative than simply discarding tied data.
- Reviewers consistently praised the paper's clarity and readability, noting it is easy to follow and explains the extensions clearly.
- The paper conducts extensive experiments across multiple real-world tasks (machine translation, summarization, mathematical reasoning) to validate the practical value of the proposed methods. It demonstrates superior performance over strong DPO baselines and provides additional analysis on reward accuracy, regularization, and convergence.
- The work directly addresses an important, under-investigated issue in preference alignment, demonstrating clear and interesting connections with established utility theory.

(c) Weaknesses
- The constructed datasets used for experiments were not publicly available, raising concerns about reimplementability.
- Hyperparameter analysis issues:
- The paper's premise that discarded "tie" data necessarily possesses high informational value was debated, with questions about potential low-quality or noisy data included as ties.
- The formulation requires annotated tied data, and it was unclear how such data could be reliably collected outside the chosen tasks, or if this annotation portion could be learned.
- Questions were raised about whether a simpler extension of DPO (e.g., with different weights for ties) could also perform well, or if the benefit strictly came from the new modeling or the extended dataset.

(d) Reasons to accept
- The paper tackles a fundamental practical limitation of DPO, which is the prevalent discarding of tie data in preference alignment. This problem is significant because ties are common in human-judged comparisons and represent valuable information.
- The introduction of DPO-RK and DPO-D provides a technically sound and novel approach to incorporate ties by building on established probabilistic models, demonstrating a deeper understanding of preference modeling beyond clear win/loss scenarios.
- Empirical demonstration of effectiveness across diverse tasks.

(e) Rebuttal:
- Reviewers questioned how tied pairs (TPs) were constructed and whether they held significant value. Authors explained four methods to construct TPs (e.g., using external metrics, conflicting annotations, or identifying pairs where both responses are correct). They provided empirical evidence that tied pairs contribute to a strong regularization effect and improved performance, supporting their utility, and committed to publishing the datasets.
- Concerns arose about the lack of novelty and the work being an incremental extension. Authors highlighted their contributions: the DPO-RK/D formulation, being the first to show the regularization effect of including ties in DPO, and proposing novel ways to model previously discarded data as ties.
- Concerns were raised about the modesty of improvements and lack of significance testing. For Machine Translation, authors stated that a >+1.4 gain in KIWI-XXL is generally considered significant. For mathematical reasoning, they ran five additional random seeds, reporting median accuracy and standard deviation, which showed consistently higher median accuracy for tie-compatible variants, with gains generally greater than one standard deviation.
- Reviewers questioned the narrowness of tasks and MT's suitability. Authors explained MT is ideal for studying ties due to its natural variability and available automatic metrics. They also showed that vanilla DPO performs poorly at classifying held-out tied pairs, while DPO-RK/D models achieved higher overall classification accuracies for ties.